# Imputation of label-free quantitative mass spectrometry-based proteomics data using self-supervised deep learning

Henry Webel [1,2], Lili Niu [1], Annelaura Bach Nielsen[1], Marie Locard-Paulet [1,3], Matthias Mann [1,4], Lars Juhl Jensen[1] & Simon Rasmussen [1,2,5] ✉

Imputation techniques provide means to replace missing measurements with a value and are used in almost all downstream analysis of mass spectrometry (MS) based proteomics data using label-free quantification (LFQ). Here we demonstrate how collaborative filtering, denoising autoencoders, and variational autoencoders can impute missing values in the context of LFQ at different levels. We applied our method, proteomics imputation modeling mass spectrometry (PIMMS), to an alcohol-related liver disease (ALD) cohort with blood plasma proteomics data available for 358 individuals. Removing 20 percent of the intensities we were able to recover 15 out of 17 significant abundant protein groups using PIMMS-VAE imputations. When analyzing the full dataset we identified 30 additional proteins (+13.2%) that were significantly differentially abundant across disease stages compared to no imputation and found that some of these were predictive of ALD progression in machine learning models. We, therefore, suggest the use of deep learning approaches for imputing missing values in MS-based proteomics on larger datasets and provide workflows for these.

Proteomics is a technology for the identification and quantification of proteins to answer a broad set of biological questions[1] and together with RNA and DNA sequencing offers a way to map the composition of biological systems. It is widely applied across many fields of research including identification of biomarkers and drug targets for diseases such as alcoholic liver disease (ALD)[2], ovarian cancer[3] and Alzheimer's disease[4]. Different workflows have been developed for analysis of body fluids, cells, frozen tissues and tissue slides, and are rapidly evolving. Recent technological advancements have enabled proteome analysis at the single cell or single cell-population level[5,6], allowing the selection of single cells using image recognition[7]. However, for most approaches, missing values are abundant due to the semi-stochastic nature of precursor selection for fragmentation and need to be replaced for at least some parts of the data analysis. Currently, imputation of missing values in proteomics data usually assumes that the protein abundance was below the instrument detection limit or the protein was absent. In general, the community differentiates between missing at random (MAR) which is assumed to affect all intensities across the dynamic range, whereas missing not at random (MNAR) becomes more prevalent the more the intensity of a peptide approaches the limit of detection of the instrument. However, not all missing values are due to this mechanism, and by assuming the limit of detection as the reason for missingness will lead to potentially wrong imputations and subsequently to biased statistical results that are limiting the conclusion from data. A strategy is therefore to combine missing completely at random (MCAR) and simulated MNAR in comparisons[8].

[1]Novo Nordisk Foundation Center for Protein Research, Faculty of Health and Medical Sciences, University of Copenhagen, Copenhagen N, Denmark. [2]Novo Nordisk Foundation Center for Basic Metabolic Research, Faculty of Health and Medical Sciences, University of Copenhagen, Copenhagen N, Denmark. [3]Institut de Pharmacologie et de Biologie Structurale (IPBS), Université de Toulouse, CNRS, Université Toulouse III - Paul Sabatier (UT3), Toulouse, France. [4]Department of Proteomics and Signal Transduction, Max Planck Institute of Biochemistry, Martinsried, Germany. [5]The Novo Nordisk Foundation Center for Genomic Mechanisms of Disease, Broad Institute of MIT and Harvard, Cambridge, MA 02142, USA. ✉e-mail: srasmuss@sund.ku.dk

Various acquisition methods have been developed including data-independent acquisition (DIA), BoxCar and PASEF to alleviate the "missing value" problem in data-dependent acquisition (DDA) methods[9–11]. Advances in informatics solutions have also greatly improved data analysis of mass spectra acquired by these acquisition methods and consequently proteome depth and data completeness[12]. However, missing value imputation of search results for downstream analysis remains a recurring task for most applications. The noise in data from the instrument as well as peptide identification is most abundant for label-free quantification proteomics in DDA with missingness ranging from 10-40%[13], but for instance, blood plasma measured using DIA in a study of ALD still contained 37% missing values across all samples and protein groups before any filtering. Independent of the proteomics setup, once data is to be analyzed, the remaining missing values between samples have to be imputed for most methods. Therefore, how they are handled will influence the downstream results.

Several methods have been evaluated for imputation of MS proteomics data with an overview and a benchmark provided in work by Wang et al., using NAguideR[14]. Despite an abundance of imputation methods, an often-used approach to impute data at the protein group level is to use random draws from a down-shifted normal (RSN) distribution. The mass spectrometry (MS) signal comes from ions and most people are interested in the summary of ions through peptide spectrum matches to groups of proteins. The protein intensities, stemming from aggregation of the precursor and/or fragment ion values in MS1 and MS2 scans, are assumed to be log-normally distributed, i.e. the log transformed intensities are entirely determined by their mean and variance. In RSN replacements are then drawn using a normal distribution with a mean shifted towards the lower detection limit with a reduced variance. This is done on the assumption that the data is left-censored, i.e. that proteins are missing due to absence or lower abundance than the instrument detection limit. Following this line of thought, several studies focus on determining what works best for different causes of missing values using some form of simulation[8,13]. Other studies focus their analysis on post-translational modifications[15], the best combination of software tools, datasets and imputation method[16], normalization and batch effects correction[17] or downstream analysis[18,19]. Other methods have been developed to handle specific missing mechanisms, for instance, random imputation, fixed value imputation such as limit of detection or x-quantile of feature, model-based imputation using k-nearest neighbor (KNN), linear models[13] or tree-based models[14]. These either impute using a global minimum, a statistic calculated on a single feature or a few features, with the need to iteratively consider each feature at a time. Finally, approaches such as DAPAR and Prostar offer several methods for imputing left-censored data, e.g. the widely used drawing from a normal distribution around the lower detection limit where the Gaussian mean and variance are estimated using quantile regression, abbreviated QRILC[8,20,21]. Their latest development is Pirat exploiting correlations of precursors from the same protein group[22]. MSStats offers end-to-end statistical analysis. It requires MaxQuant running samples jointly and providing both precursor as well as protein groups next to grouping information and does not impute all values.

Most previous work on developing methods for imputation of MS proteomics data focus on small scale setups where they for instance evaluate two separate groups in three replicates[16,23]. Other studies prefer to reduce the number of initial missing values by transferring identification from one run to the next using e.g. Match Between Run implemented in MaxQuant or cross-assignment by Proline[24,25]. Alternatively, an established laboratory method to tackle variability in small-scale setups is to use replicates of samples to have complementary measurements, in order to transfer identifications between runs[26]. The specific evaluation strategy varies between the setup of the data and the missing values simulation approach, but to

our knowledge, no scalable workflows are provided to run the evaluation on a new single tabular dataset generating a validation and test dataset, as well as a generic and flexible approach to imputation.

We turn to machine learning for imputation of missing intensities as it offers the possibility to learn from the data itself. Deep Learning (DL) has been used to improve over existing machine learning models in a variety of biological data problems[27–29]. DL has been successfully applied to predict peptide features such as retention times, collisional cross-sections and tandem mass spectra, significantly boosting the peptide identifications and precision of searches of MS-based proteomics[30–34]. We use intensities from search results generated from tools like e.g. MaxQuant or Spectronaut and apply DL methods to impute these. We considered three types of models that process the search results of precursors, peptides or protein groups slightly differently. First, we considered a collaborative filtering (CF) approach, where each feature and each sample is assigned a trainable embedding. Second, we considered an autoencoder with a deterministic latent representation - a denoising autoencoder (DAE). Third, we considered a variational autoencoder (VAE) as a generative model that encodes a stochastic latent representation, i.e. a high-dimensional Gaussian distribution. Although the inputs to all models are intensities for a feature and sample combination, the training objectives, complexity, and therefore capabilities of the models are different. The CF and autoencoder objective only focus on reconstruction, whereas the VAE adds a constraint on the latent representation. Furthermore, the first two modeling approaches use a mean-squared error (MSE) reconstruction loss, whereas the VAE uses a probabilistic loss to assess the reconstruction error.

Here, we use large ($N \approx 450$) and smaller ($N \approx 50$) MS-based proteomics datasets of HeLa cell line tryptic lysates acquired on a single machine (Q Executive HF-X Orbitrap) over a period of roughly two years to evaluate the general performance on repeated measurements of an homogenous biological sample on a medium to large-scale dataset. We apply three different DL models (CF, DAE, VAE) which create feature representation holistic for the entire distribution in a given dataset prior to any normalization. For evaluation, we develop a workflow that allows comparison between the three DL models and 27 approaches (Supplementary Table 1). Finally, we apply the VAE to a study of ALD patients and identify 30 (+13.2%) more significantly differential abundant protein groups in comparison to no imputation and that additional protein groups can be leveraged for predicting disease. We name our set of models and workflows proteomics imputation modeling mass spectrometry (PIMMS) and make the workflows, code, and example configs available at https://github.com/RasmussenLab/pimms. To enable reproducibility and adaptation to new data and strategies, we share our Python code along snakemake workflows.

## Results

### Evaluating self-supervised models for imputation of MS data
We assessed the capability of three unsupervised models for proteomics data imputation. First, we considered modeling proteomics data using CF assigning each sample and each feature an embedding vector and using their combination to predict intensity values. Second, we considered a standard autoencoder, training it using a denoising strategy that has to learn to reconstruct masked values making it a DAE. Third, we applied a VAE with a stochastic latent space (Fig. 1a). The two autoencoder architectures used all features to represent a sample in a low dimensional space, which was used to reconstruct the original data. In contrast, the CF model had to learn both a latent embedding space for the samples and features. We compared these to a heuristic-based approach of median imputation per feature across samples and 26 other approaches such as k-nearest neighbors (KNN) or random forest (RF) (Supplementary Table 1). While the DL methods, KNN and median imputation were

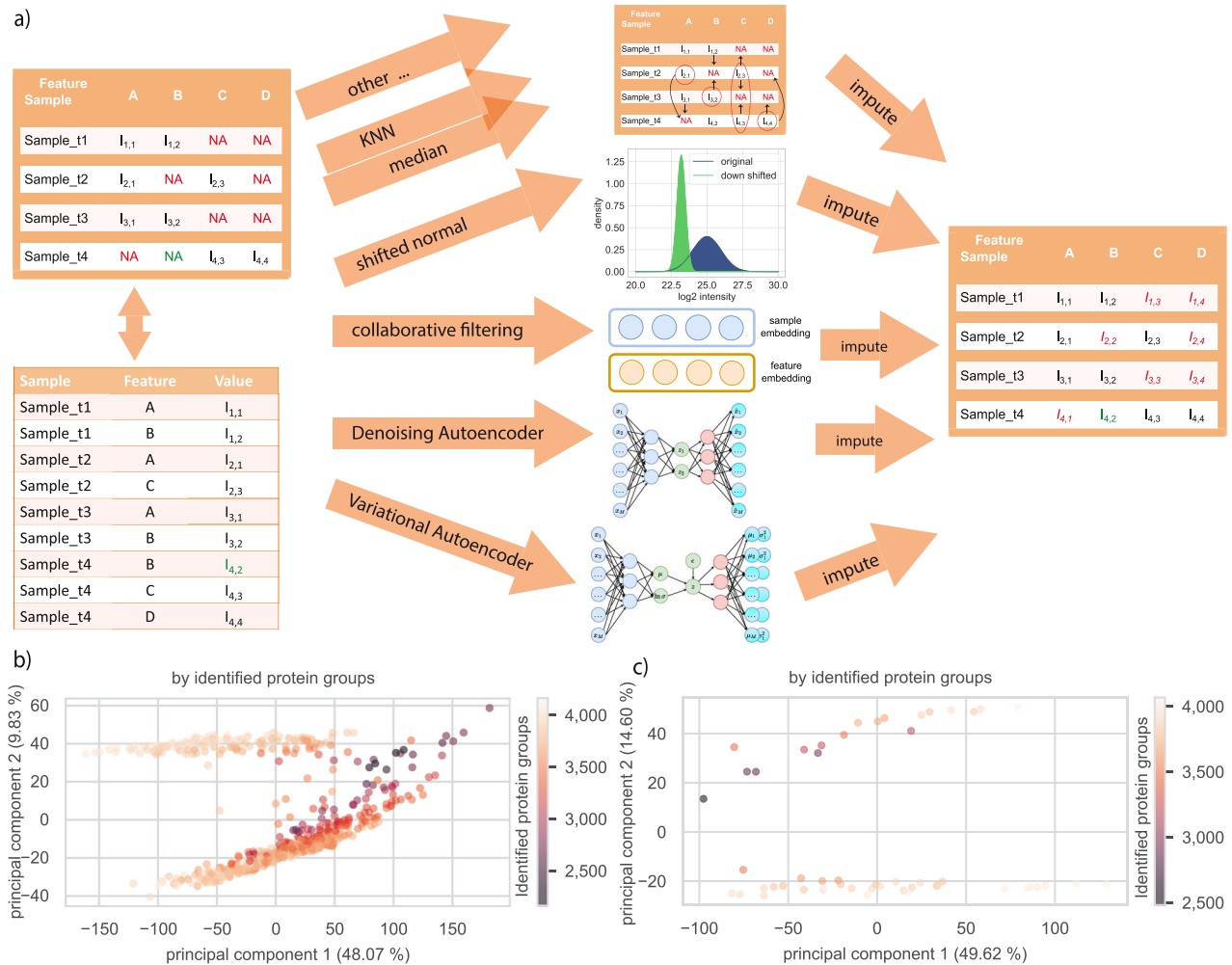

**Fig. 1 | Overview of workflow for downstream analysis tasks and HeLa dataset.**
**a** Single tabular results taken from MS data analysis software (search and quanti-fication) were used as input for downstream analysis. Here we used MaxQuant for data dependent acquisition to analyze raw MS data. We compared three different self-supervised DL approaches with 27 other methods: median imputation and KNN interpolation exemplified. Green and red not-available (NA) indicate simulated and real missing values. **b** Principal component one versus two of 539 selected HeLa runs for protein groups recorded on one instrument. **c** Same as (**b**), based on the 50 runs forming the small development dataset. We used a cutoff of 25% feature prevalence across samples to be included into the workflow shown in (**a**). Samples were filtered in a second step by their completeness of the selected features (Supplementary Fig. 1).

able to also impute large datasets, nine methods implemented in R failed to scale to high-dimensional data.

Our development dataset consisted of 564 HeLa runs of one Q Executive HF-X Orbitrap generated during continuous quality control of the mass spectrometers[35]. We initially investigated the structure of the dataset using the first two principal components (Fig. 1b), which grouped the samples into two separate clusters. The median prevalence per protein group, i.e. the median number of samples where a protein group was detected, was 526 samples [min: 45, max: 564], and samples had a median of 3768 protein groups [min: 2170, max: 4185]. We generated another smaller development dataset using the most recent 50 samples (Fig. 1c) to test dependence of performance on sample size. Whereas median imputation per feature across samples did not condition their imputation on the value of other features in a given sample, our and other machine learning approaches consider other feature values. Validation and test data were drawn with 25, 50 or 75 per-cent MNAR using the procedure laid out in Lazar et al.[8] from all samples in a dataset. This ensures that lower intensities are represented sufficiently, but also makes training harder due to additionally removing values from low abundant features with fewer quantifications.

## Imputing precursors, aggregated peptides, and protein group data

We applied the imputation methods to the development dataset, e.g. consisting of 564 samples for protein groups using our selection cri-teria (Supplementary Fig. 1). We ran several configurations using a grid search to find the best configurations using simulated missing values in a validation and test split from all samples (see Methods, Supple-mentary Data 1, 2). In absolute numbers these were 100,001 for the protein group, 616,561 aggregated peptides and 661,817 precursors intensities in the test set for our development dataset of instrument 6070. When investigating the performance of the imputation methods we used the mean absolute error (MAE) on the log2 scaled intensities between predicted and true measured intensity values on our simu-lated missing values. Focusing on the best performing models from the grid search, we evaluated these in a setup of 25 percent MNAR in the simulated missing data. We found that the DL approaches had less than half MAE compared (0.55, 0.54 and 0.58 for CF, DAE and VAE, respectively) to the median imputation with MAE of 1.24. KNN of samples across HeLa cell line measurements had a MAE of 0.59 using the scikit learn implementation and of 0.68 using an R based imple-mentation using impute[36,37]. The R based random forest (RF) imputation[38] only completed for the protein groups and had an MAE of

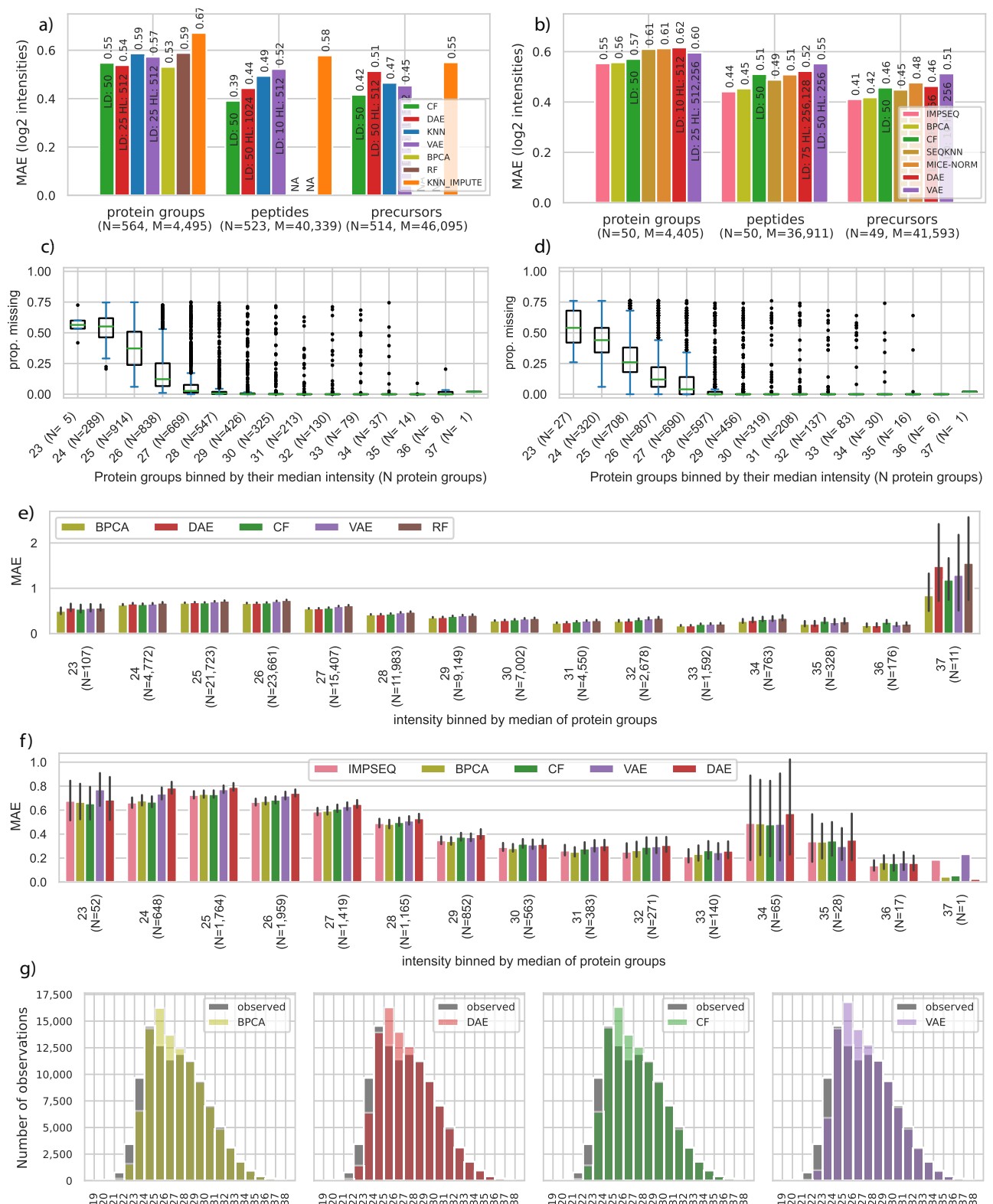

0.59 and a Bayesian principal component analysis (BPCA) with 0.53. Thus, BPCA was slightly better (0.01 MAE) compared to the second-best model on protein groups however it did not finish for the other two datasets within 24 h. Overall, when comparing the performance across levels of data aggregation for the best models (Fig. 2a,c, Supplementary Data 3) we found that the self-supervised models performed similarly to other methods. Across all models, we found the overall performance to be the worst for the protein-level data, better

for aggregated peptides, and best for precursors. This is in line with previous results of Lazar and co-workers that showed better performance for lower levels of aggregations as it avoids implicit imputations by a neutral element[8]. A higher share (25, 50 and 75 percent) of MNAR simulated missing values was associated with a decreased overall performance of most methods (Supplementary Data 3). One reason for this could be that features with fewer observations are oversampled and the total number of available training data points drops as low as

**Fig. 2 | Different levels of MS-based proteomics data can be imputed using self-supervised DL models. a** Performance of best imputation methods at the level of protein groups, aggregated peptides, and precursors for MaxQuant outputs with 25 percent MNAR. Mean absolute error (MAE) is shown on the y-axis. Blue: KNN (scikit-learn), green: CF, red: DAE, purple: VAE, olive: BPCA, brown: random forest (missForest), orange: KNN_IMPUTE (impute), pink: IMPSEQ, dark-orange: MICE-NORM, sand: SEQKNN. The DL methods performed better or equally good in comparison to other models. Performance was similar for all three DL models on each data level and less aggregated data, i.e models on precursors and aggregated peptides performed better compared to models on protein groups. BPCA and RF did not finish for aggregated peptides and precursors within 24 h. Models were ordered by overall performance on the three datasets combining the best five for each. N: number of samples, M: number of features. **b** As (**a**) but showing a decrease in performance for a subset of the data with a maximum of 50 samples. The CF adapted better to smaller sample sizes. **c, d** Protein groups medians were binned by their integer median value and the boxplot of the proportion of missing values per protein group is shown for large and small development. N: Number of protein

groups in bin in parentheses. The box of the boxplots extends from the first quartile to the third quartile of the proportions with their median as a separate line. The whiskers extend no more than one and a half times the interquartile range, ending at the farthest proportion within that interval. Outliers are plotted as separate dots. **e** MAE 95% bootstrapped confidence interval for protein groups intensities in the test split of the larger development dataset binned by the integer value of the protein group's median intensity in the training data split (N: number of intensities in a bin from test split, 4495 protein groups, 103,902 intensities, models ordered by overall performance, 1000 draws to compute the confidence interval). Models on protein groups with a median intensity above 26 performed in the range of the overall performance shown in (**a**). **f** MAE 95% bootstrapped confidence interval for protein groups in test split of smaller development dataset binned by integer value of intensity (N: number of intensities in a bin from test split, 4405 protein groups, 9327 intensities, models ordered by overall performance, 1000 draws to compute the confidence interval). **g** Imputed and ground truth (observed) intensities for test data on large protein group development dataset for top four overall models.

four intensities in the training data split. To balance available data for low abundant features and statistical power in evaluation, we set the share of MNAR to 25 percent in our evaluations.

Runtime for imputation of protein groups including data loading, manipulations and training were between 14 s using the minimum and roughly 8 h for IRM (for the best models in h:min:sec: KNN-IMPUTE: 0:19, KNN: 0:41, VAE: 1:58, DAE: 2:24, CF: 2:28, RF: 1:05:02, BPCA: 6:37:56) using 1 CPU and up to 192 GB of shared memory for the PIMMS models (Supplementary Data 4). Runtimes of PIMMS models for the precursor dataset were higher (KNN: 4:17, DAE: 33:08, VAE: 1:35:51 and CF: 18:23), but could run faster with special hardware (GPUs). Among the models with good performance we found that BPCA, MICE-NORM, SEQKNN and RF did not scale to the high dimensional datasets of peptides and precursors for the large datasets. IMPSEQ ran into errors for all larger datasets. In general runtimes varied based on the number of epochs, mini batch-size, model architecture, patience for early stopping and the initial random weights. Additionally, the grid search results showed that the models could be trained without prior normalization and were able to fit the data using many different hyperparameter configurations (Supplementary Fig. 2). Reducing the training sample by roughly a tenth, performance on the smaller development dataset (N = 50) was comparable to the performance on the larger development dataset, while runtime reduced by 5 to 15 times for precursors (Median: 0:23, KNN: 0:33, VAE: 1:04, DAE: 0:52, CF: 1:06). On the medium sized HeLa datasets all models completed (Supplementary Data 4) for the protein groups and 26 out of 27 for the peptides and precursors. We found that IMPSEQ[39] and BPCA[40] performed best on the 50 homogenous HeLa samples although some with longer runtimes compared to PIMMS models (e.g. on precursors: IMPSEQ: 1:09, BPAC: 20:44, MICE-NORM: 4:42). For fewer than 50 samples the PIMMS models can fit the data, but alternatives were better suited (Supplementary Data 3). In summary, this indicated that on an unnormalized, intensity varying dataset (Supplementary Fig. 3a) from a single machine the PIMMS models were able to capture patterns between detected features to impute values for as few as 50 samples with competitive performance to other state-of-the-art methods.

### Imputation was consistent across a wide range of intensities

We evaluated the performance across the dynamic range of intensities by binning test intensities by their feature's training median and reporting the average error per bin for the best five models. For protein groups on both development datasets the minimal median intensity was 23 for protein groups (Fig. 2c, d). The average error for intensities was roughly twice as large as for intensities from the minimum median intensity bin in comparison to the best performance (Fig. 2e, f). Interestingly, the worst performance on the large HeLa dataset was observed for protein groups with the highest observed median

intensity, which for intensities from one protein group had a two to four times worse MAE than the overall MAE. The relative performance between the models was consistent across the bins. If we compared this to the relative distribution of imputed intensities (with errors) to the unmodified intensities in the test split, we found that low intensity values were partly imputed towards the center of the intensity distribution (Fig. 2g). Furthermore, we investigated whether there was a difference in the accuracy of the imputation based on how often a protein group was observed. Here we found that the MAE varied between 0.6 and 0.8 for proteins observed in 25-80% of the samples, whereas for proteins observed in more than 80% of the samples the MAE decreased to below 0.4 (Fig. 2c, e). We observed a similar trend when analyzing the smaller dataset of only 50 samples (Fig. 2d, f). This indicated that some protein groups were harder for all the methods to impute than others, but also that the CF, DAE, and VAE predicted consistently across a wide range of protein groups intensities. We found similar results for the two other levels of data, aggregated peptides and precursors (Supplementary Fig. 4). Finally, we investigated how stable the self-supervised models trained. In a first step, we randomly permuted all protein groups of the large development dataset, which leaves the median performance unchanged. Accordingly, training models on randomly permuted data could not outperform the imputation by the median of the training data split with an MAE of 1.25 (Supplementary Fig. 5a, Supplementary Data 5, CF: 1.24, VAE: 1.26, KNN_IMPUTE: 1,30 DAE: 1.32). Next, we trained the self-supervised models five times on the same data split as well as repeated training five times on new splits. The performance between fitted models varied in a narrow margin (protein groups: CF: 0.538-0.550, DAE: 0.535-0.545, VAE: 0.561-0.587 [Min-Max]) (Supplementary Figs. 5b,c, Supplementary Data 5). We therefore conclude that the DL models could be consistently fitted to data and that self-supervised models were able to fit the data holistically for imputation purposes.

### Competitive within-sample and feature-wise between-sample correlation

We evaluated performance without a specific distance measure by evaluating Pearson correlations of simulated missing values to the truth (Supplementary Fig. 6, Supplementary Data 6). The mean Pearson correlation between samples for protein groups was a bit higher for BPCA than the others, including the self-supervised models for the imputations on the large development dataset (BPCA, DAE: 0.86, CF: 0.85, VAE: 0.84, RF: 0.83, KNN: 0.82, KNN_IMPUTE: 0.77). The correlation between features within a sample was higher in general, with a mean correlation of around 0.95 for all models seen among the best along the sorting by the MAE (BPCA, DAE, CF: 0.96, VAE, RF, KNN: 0.95, KNN_IMPUTE: 0.93). This showed that the ordering within a sample was better than the correlation of protein groups between samples as

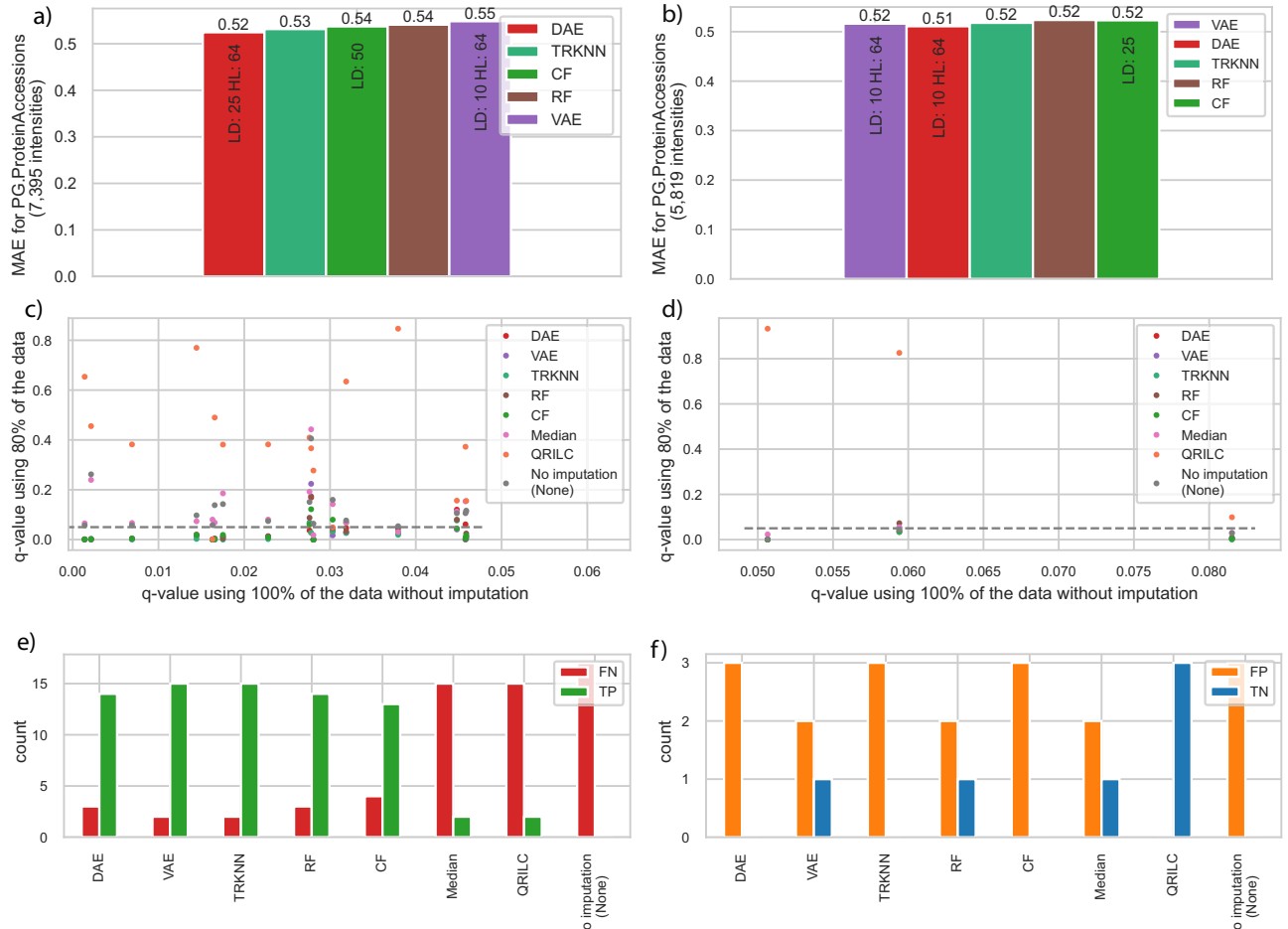

**Fig. 3 | Performance on simulated missing data with 25 percent MNAR in ALD dataset and effects of imputation on differential analysis. a** Performance for protein groups of plasma proteome data using a share of 25 percent MNAR simulated missing values on the full dataset and (**b**) on the 80% dataset using the same configurations (LD: latent dimension, HL: hidden layer dimension). The performance of best five imputation methods for protein groups of plasma proteome data using a share of 25 percent MNAR simulated missing values (red: DAE, darkgreen: truncated KNN (TRKNN), green: CF, brown: RF, purple: VAE) **c** q-values, i.e. multiple testing adjusted p-values using Benjamini-Hochberg method, for 17 protein groups which were differentially abundant using the full dataset without imputation, but not for the 80% dataset without imputation. The gray line indicates the five percent FDR cutoff. No imputation (None) shows the result without

imputing values. Original *p* values were calculated using an analysis of covariance (ANCOVA), i.e. a regression of the protein intensity along several clinical covariates. **d** q-values for three protein groups which were differentially abundant using the 80% dataset without imputation, but not initially when using all data. **e** Count of false negative (FN) and true positives (TP) per method on the reduced dataset taking the 17 differentially abundant protein groups from (**c**) on the full dataset as ground truth. **f** Same as in (**e**) for three protein groups in (**d**), labeling differentially abundant protein groups in the 80% dataset as false positives (FP) since they did not show up as differentially abundant in the full dataset. The three examples are around the FDR cutoff without imputation. True negatives (TN) are not differentially abundant here (Supplementary Data 9). No imputation (None) is the reference defining the labels.

the overall abundance level of single protein groups vary across samples. For the smaller development dataset we found similar trends of within-sample correlation (IMPSEQ: 0.80, BPCA: 0.80, CF: 0.79, SEQKNN, VAE: 0.78, DAE: 0.76) and between feature sample correlation (IMPSEQ, BPCA, CF, SEQKNN, VAE: 0.95, MICE-NORM, DAE: 0.94). Both correlation comparisons, therefore, indicated that the three self-supervised models were able to model the data well without prior normalization of the data.

### Performance of PIMMS on simulated missing values in real use case

To assess the impact of imputation on a large real-world DIA dataset, we applied PIMMS to 455 blood plasma proteomics samples from a cohort of ALD patients and healthy controls[2]. After imputation we again compared how well PIMMS imputed simulated missing values with a share of 25 percent MNAR in the ALD data and found that DAE, TRKNN, CF, RF and VAE achieved similar results of MAE between 0.52-0.55 (Figs. 3a, 4a–c, Supplementary Data 3). BPCA which performed

well on the development dataset, was worse with an MAE of 2.95 and took more than 10 minutes to complete compared to between 30 seconds up to just above one minute for the best models (Supplementary Data 3, 4). In the original study RSN was used for imputation on a per sample basis, i.e. using the mean and standard deviation of all protein groups in a sample. This yielded 8 times worse results on the simulated data (Supplementary Data 7). Having many samples available, we also tested to only simulate missing values on a stratified subset of samples (by kleiner score, see "Methods") leading to 68 samples and 69 samples assigned to the validation and test split. Sampling simulated missing values only from these led to fewer available intensities for comparison but matched the results when sampling missing values from all available samples (Supplementary Data 8). The correlation across samples for each protein group was lower for the ALD data in comparison to the heterogeneous measurements of HeLa cell line data for the self-supervised models (median; DAE: 0.60, TRKNN: 0.58, CF: 0.54, VAE: 0.53, RF: 0.52) based on 377 protein groups with at least 3 intensities in test data split

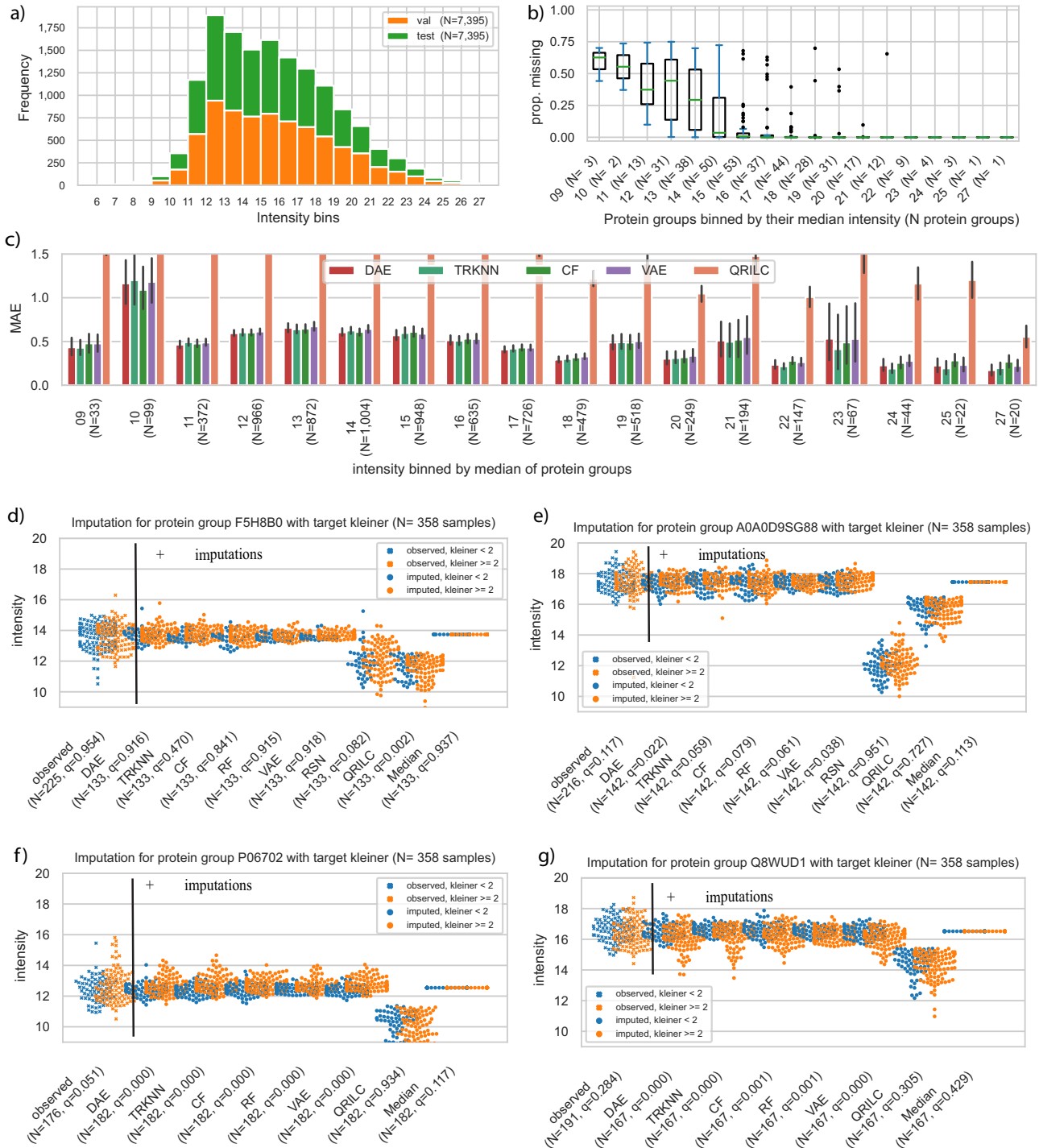

**Fig. 4 | Underlying data distribution and performance in detail on an ALD dataset. a** Stacked histogram of validation and test split using a share of 25 percent MNAR. Bins were created using the integer value of the log2 transformed intensities. N is the number of intensities in the split, each corresponds to 5 percent of all available quantifications. **b** Protein groups medians were binned by their integer median value and the boxplot of the proportion of missing values is shown. N: Number of protein groups in bin in parentheses. The box of the boxplots extends from the first quartile to the third quartile of the proportions with their median as a separate line. The whiskers extend no more than one and a half times the interquartile range, ending at the farthest proportion within that interval. Outliers are plotted as separate dots. **c** MAE with 95% bootstrapped confidence interval for protein groups intensities in the test split binned by the integer value of the protein group's median intensity in the training data split. RF was replaced with QRILC to showcase a MNAR focused method. (N: number of intensities in a bin from test split, 377 protein groups, 7935 intensities, models ordered by overall performance, 1000 draws to compute the confidence interval). **d–g** Comparison of q-values, i.e. multiple testing adjusted p-values using Benjamini-Hochberg method, based on no imputation (measured) and adding imputations using the models for four protein groups. orange: kleiner score > 1, blue: kleiner score <2, crosses in the first column indicate observed values, and dots in other imputation by models. Original *p* values were calculated using an analysis of covariance (ANCOVA), i.e. a regression of the protein intensity along several clinical covariates, either only for the observed intensities or using the imputed intensities in addition to the observed intensities.

(Supplementary Data 7). The median correlation within samples (Supplementary Data 7) was around 0.98 for all models keeping the order by MAE. This, thus, matched the overall results from the HeLa data analysis.

### Recovering lost differentially abundant protein groups

Then we tested the ability of the self-supervised methods to recover differentially abundant protein groups. To achieve this, we subset the dataset at a share of 25 percent MNAR and only kept 80 percent and compared differentially abundant proteins to using the full dataset. We found that the MAE performance on the simulated missing values with a share of 25 percent MNAR data were nearly the same compared to the full dataset (Fig. 3a, b, Supplementary Data 3). This can be explained by the relatively stable measurements of the dataset (Supplementary Fig. 3b) and indicates that most protein groups were stable across patients. Comparing the results of the differential abundance analysis using the complete dataset compared to the reduced dataset using no imputations, we observed a decrease in the number of differentially abundant protein groups from 226 to 212 (ANCOVA, Benjamini-Hochberg multiple testing correction, $p$ value $\leq 0.05$). Interestingly, we found that besides 17 protein groups losing their differential abundance, three became differentially abundant by randomly removing data. To assess the effectiveness of various imputation methods, we used no imputation as a benchmark. We found that VAE and TRKNN imputation techniques successfully recovered 15 out of the 17 protein groups that lost their differential abundance (DAE, RF: 14, CF: 13, Median, QRILC: 2) (Fig. 3c, e, Supplementary Data 9). Three protein groups became differentially abundant without imputation when intensities were removed in the 80% dataset, which we labeled as false positives to follow our approach (Fig. 3d, f). However, the three protein groups were nearly statistically significantly abundant in the 100% dataset and therefore adding back patients through imputation to the analysis can be seen as beneficial. This indicated that the PIMMS models could recover lost signals with simulated missing data with a share of 25 MNAR.

### More differentially abundant proteins when using PIMMS

Then we investigated the number of differentially abundant protein groups as well as the ability of the plasma proteome to predict the fibrosis status of 358 individuals for the top five models, RSN, QRILC and median imputation (Supplementary Data 10). The fibrosis status was based on a liver biopsy, and individuals with all fibrosis scores ranging from zero to four were included. By using these models for imputation of missing data, leaving the available data as it was, we were able to perform the analysis for 377 protein groups (ANCOVA, Benjamini-Hochberg multiple testing correction, $p$ value $\leq 0.05$) compared to only 313 protein groups when using the RSN imputation approach as originally applied in Niu et al.[2]. Comparing RSN and VAE we observed that both methods replaced missing values with a distribution shifted towards the lower abundance region. However, the maximum intensity for missing values was higher for the VAE with a value of 21 compared to the RSN with a value of 15 for protein groups (Supplementary Fig. 7). Translating the overall shift in distribution by the VAE to the RSN idea, the intensity distribution was shifted by one standard deviation and the variance was shrunk by 0.7 in comparison to 1.6 standard deviations and 0.3 shrinkage in the ALD-RSN setup. This difference underlies a fundamental difference in the approaches and can be generalized for all other models (Supplementary Fig. 7). Whereas the RSN always assumes missing values due to low abundance, the VAE and other approaches assign some missing values a higher intensity if other protein groups in the same sample suggest that the missing value occurred rather due to a missed detection than low abundance. QRILC is a more nuanced MNAR method that does not only assign low intensities for missing values and we therefore added it to the comparison, although it shifts the distribution more strongly than RSN (Supplementary Fig. 7).

When performing differential analysis, we found that 209 of the 313 protein groups originally included were significantly differentially expressed using the RSN approach. However, 212 were significant without any imputation and up to 221 were significant using the other methods (TRKNN: 221, VAE: 220, DAE, CF: 219, RF: 217, QRILC: 216, Median: 211). Adding the newly included 64 protein groups, 258 of 377 in the PIMMS-VAE setup were differentially expressed, an increase by 23% to the originally included and imputed protein groups with RSN (see Methods and Supplementary Data 11). We found that using the nine approaches, 287 decisions from the differential expression analysis were the same for the 313 shared protein groups. We went on analyzing the 26 differences in detail (Supplementary Data 12). First, we found that nine of the 26 protein groups were differentially abundant without imputation. For these nine protein groups RSN imputation led to non-significant results, whereas the top five models found these nine to be significant. For the four least significant protein groups without imputation of missing values QRILC imputation made these significantly abundant. For three protein groups with multiple testing adjusted p-values (q-values) just above the 0.05 threshold the self-supervised models and TRKNN imputation led to differential abundance, whereas only Median imputation did not.

On one hand when using QRILC imputations six protein groups (*IGLV9-49, TTN, DEFA1B/DEFA3, PRDX1, ACTA2/ACTG2, SSP2, F7*) were significant which none of the other non-MNAR methods found, with few having very diverging q-values. Few relatively non-prevalent protein groups, i.e. here below 70% prevalence (250/358 and less) of quantified samples, showed a strong difference in differential analysis testing between e.g. the VAE and the RSN imputation. The relatively non-prevalent (222/358) protein group F5H8B0/P08709/P08709-2 associated with the gene *F7* was clearly not significant using the VAE or no imputation, but passed the statistical thresholds when using QRILC imputation and leaned towards significance using RSN. On the other hand, a relatively rarely quantified protein A0A0D9SG88 (gene *CFH*) was significant using VAE imputation, but not RSN or QRILC (Table 1, Supplementary Data 12). Visualization using swarm plots indicated that the RSN and QRILC imputation for these two protein groups were shifted downwards of the original center (Fig. 4d-e). In combination with a slight imbalance of targets in the 142 imputed samples, this led to different results. Similarly, for two newly included protein groups P06702 (gene *S100A9*) and Q8WUD1/Q8WUD1-2 (gene *RAB2B*) imputation by a fixed median value or QRILC did not lead to significant results (Fig. 4f-g, Supplementary Data 12). It could, therefore, be good scientific practice to check the effect of imputation on differential analysis before a single protein group is selected for further analysis, especially for explicitly random methods such as RSN.

Finally, expanding the analysis to the 64 additional protein groups included here, we found 16 of them to be significantly differentially abundant without any imputation and 38 using VAE imputations (TRKNN: 44, VAE: 38, RF: 34, CF: 33, DAE: 32, QRILC: 19, Median: 14, see Supplementary Data 11). Therefore, apart from the difference from the shared protein groups, which yielded 11 more significant hits, imputation using e.g. the VAE allowed us to identify 38 additional protein groups that were not considered for statistical analysis in the original study.

### Robustness of differently abundant protein group identification

Next, we decided to analyze in more detail which protein groups were differentially abundant using the different models and how reliable the differential outcome was per model. We repeated the analysis on the ALD data ten times with 25 percent MNAR simulated missing values, re-training the models on the same training data split. In this scenario median, TRKNN and RSN imputations were guaranteed to yield the same result for all repetitions. Of the 313 originally included protein groups, 27 did not have the same differently abundant outcome for all models on all ten repetitions. Especially for protein groups with few

**Table 1 | Examples of diverging decisions between imputation methods**

| Protein Group | F5H8B0 P08709 P08709-2 | A0A0D9SG88 | A0A075B6R2 | P02741 | I3L0A1 J3KPA1 P54108 P54108-2 P54108-3 | P59665 P59666 |
|---|---|---|---|---|---|---|
| Gene | F7 | CFH | IGHV4-4 | CRP | CRISP3 | DEFA1B DEFA3 |
| None (ref.) | 0.954 | 0.117 | 0.000 | 0.030 | 0.018 | 0.365 |
| DAE | 0.916 | 0.022 | 0.000 | 0.038 | 0.001 | 0.025 |
| TRKNN | 0.470 | 0.059 | 0.000 | 0.029 | 0.000 | 0.082 |
| VAE | 0.918 | 0.038 | 0.000 | 0.039 | 0.000 | 0.023 |
| RF | 0.915 | 0.061 | 0.000 | 0.040 | 0.000 | 0.061 |
| CF | 0.841 | 0.079 | 0.000 | 0.031 | 0.001 | 0.022 |
| Median | 0.937 | 0.113 | 0.000 | 0.030 | 0.051 | 0.373 |
| RSN | 0.082 | 0.951 | 0.080 | 0.052 | 0.745 | 0.012 |
| QRILC | 0.002 | 0.727 | 0.003 | 0.062 | 0.626 | 0.029 |

Based on comparison of $q$-values, i.e. multiple testing corrected $p$-values using the Benjamini-Hochberg method, for previously included protein groups. Analysis without imputed values is denoted as None and used as reference (ref.). All diverging decisions are in Supplementary Data 11,12.

missing values, seeing the same outcome is not unexpected. However, for nine protein groups with two to 29 percent missing values, adding RSN imputations changed the result from being differentially abundant without imputation to being not differentially abundant. The results did not change running DAE, TRKNN, CF, RF and VAE ten times for these nine protein groups. A total of 46 out of the 64 newly included protein groups with a higher percentage of missing values were at least once differentially abundant by using one of the methods. Of these 46 newly included ones, 16 were differentially abundant without imputation, which except by median or QRILC imputation was not changed using the top five model imputations. Additionally, 13 were at least identified seven times as differentially abundant using DAE, TRKNN, CF, RF or VAE imputations (Supplementary Table 2, Supplementary Data 13). Using CF imputation, we found 17 protein groups to be differentially abundant in all ten repetitions. Deterministic TRKNN imputation in our setup found 28 protein groups to be differential abundant ten times. However, ten of these were only found using TRKNN imputations at least seven times (Supplementary Table 2). We conclude that there is some variability in the analysis due to imputations and that for certain imputed protein groups the choice of the imputation method is crucial.

### Novel protein groups could be biological relevant
We then investigated if these protein groups could be associated with disease using the DISEASES database[41] and found that 39 of the 64 novel proteins had an association entry to fibrosis. Of the 38 at least once newly significant protein groups 30 had an association entry to fibrosis, with four having a confidence score greater than two (Supplementary Data 13, 14). For example, the protein P05362 from gene *ICAM1* had the highest disease-association with a score of 3.3. It was found to be significantly dysregulated in the liver data and missing in the plasma data in the original study. Following this reasoning, the second highest scoring protein group was composed of P01033 and Q5H9A7 (gene *TIMP1*). This indicated that the novel protein groups using the PIMMS criteria for inclusion could be biologically relevant.

### The additionally added protein groups were predictive of fibrosis
Finally, in the work by Niu et al., the authors trained machine learning models to predict clinical endpoints such as fibrosis from the plasma protein groups. To assess the impact on the machine learning model, a logistic regression, we used the data from the differential analysis above. We replicated the workflow performed by Niu et al., and

evaluated the model using the ALD cohort for individuals with histology-based fibrosis staging data available ($N = 358$). Using minimum redundancy, maximum relevance (MRMR) approach we selected the most predictive set of features of each subset of features on the assigned training samples[42]. Using median imputation for all available protein groups yielded the best area under the receiver operating curve (AUROC) of 0.90 compared to 0.86–0.90 for DAE, TRKNN, CF, RF, QRILC and VAE on the test samples (Supplementary Fig. 8). For models trained on only the newly included protein groups the DAE and VAE model imputation led to the best AUROC using only one protein group (*TIMP1*) previously mentioned, whereas the model using five median imputed protein groups was worse (VAE: 0.80, DAE: 0.82, CF: 0.73, RF: 0.79, Median: 0.66, TRKNN: 0.76, QRILC: 0.74). Therefore, this suggests that the additional protein groups were retained with more missing values when using the PIMMS approach compared to the original study's approach.

### Discussion
Imputation is an essential step for many analysis types in proteomics, which is often done heuristically. Here we tested three models using a more holistic approach to imputation. We showed that CF, DAE and VAE models reached a similar performance on simulated missing values across the entire distribution of the data - including low abundant features. In comparison to most other methods they scaled better to high dimensional data or outperformed fast implementations as scikit-learn based KNN, which was especially beneficial when working with peptides instead of protein groups to avoid implicit imputations[8]. Further, we investigated the effect of the imputation method on a concrete analysis, using DIA data from 358 liver patients. Here we found that missing values were imputed by the models towards the lower end of the distribution but less pronounced as when using QRILC or RSN imputation which shifts all replacements towards the limit of detection (LOD) in a sample. We believe that this is due to the lowest abundant features limiting the learned data distributions and that some features are not set towards the LOD by the model due to being missing at random. We therefore argue that our holistic model-based imputations are more conservative than e.g. the RSN imputations and that the three self-supervised DL models offer a sensible approach to proteomics imputation while scaling well to high feature dimensions. We found that all methods besides RSN were a better choice for the ALD dataset analyzed.

Simulating 20 percent missing values with a share of 25 percent MNAR we saw the ability to recover signal by the three semi-supervised

models. TRKNN and RF recovered lost signal in this application, but median imputation and QRILC were too heuristic to recover most of the signal (Fig. 3). However, the analysis also revealed that some protein groups will be false positives or negatives as removing some observations can change the outcome. This suggests that imputation using our or other data driven models can recover lost biological signals which is in line with the observation that significant values were not made insignificant when using data driven models (DAE, TRKNN, CF, RF, VAE) in comparison to RSN or QRILC on the ALD cohort (Fig. 4). Finally, we offer a workflow to reproduce the comparison done here for all 30 general imputation methods using any other single tabular data provided by the user.

A limitation of the model-based approach is that the models should only be used for imputation if the samples are related. Therefore, the best imputation strategy will be dependent on the experimental setup. We showed that the models can learn to perform imputation on plasma samples from a diverse set of clinical phenotypes ranging from healthy to liver cirrhosis. However, the data-driven models would not perform well when imputing for instance one liver proteome together with ten plasma proteomes. In such a case the data-driven models would not have any other liver proteomes to learn from. If replicas on a small number of samples are the study design, KNN interpolation can be a good remedy or using a set of tests which among others capture missingness[23]. DAE and VAE are not suited to be trained on too few samples, however the exact cutoff will need to be evaluated on a per dataset basis. CF will by the training design not be too dependent on the number of samples if trained sufficiently. We showed that performance is at least competitive with at least 50 samples. In summary, for highly varying features in a complex experimental setting with many differing samples, a holistic model trained with all features and potentially additional related samples can capture dependencies between features - such as the ones implemented in PIMMS.

In general, the modeling approaches here are restricted to the samples in a particular study and all models are fitted for each new dataset. However, transfer of models between datasets can be envisioned although a recent study suggested that this brings no benefits[43]. The potential to fine-tune a model trained on one dataset to a new dataset for a fixed set of features is possible without further efforts for autoencoders. For CF one would need to find the closest training samples in the case where samples are separated strictly into train, validation and test set. However, feature embeddings could be transferred and extended easily. Therefore, all models could potentially be envisioned in a clinical setup, where models are re-trained with the latest samples. This could be implemented using similar cohorts, e.g. for the same tissue and similar patients, which is then the basis to build a database of tissue specific models - or by incorporating tissue embeddings as an additional source of information. The difficulty in achieving this will be a stable setup for comparable results without major batch effects due to sample handling or different instruments. How to approach and the potential for data integration from different setups is an unresolved issue. In general, community-curated benchmarks including datasets and detailed metrics should be discussed by creating one or more ProteoBench modules in a community effort[44].

To ensure reproducibility and further extension we offer an evaluation workflow for simulated MCAR missing values of the entire data distribution and oversampling low abundant intensities (MNAR) instead of only reporting results on our specific datasets. Everything is available and continuously tested on GitHub, including the workflows, which allows for additional methods to be added to our comparisons. This includes comparisons on simulated missing values with varying degrees of MNAR which can be extended to further holistic models. The potential extensibility of the workflow allows for comparison of different ideas on different datasets, including the downstream analysis.

We evaluated imputation on different levels of proteomics features and found that lower-level data was easier to learn due to being less aggregated[8]. Therefore, it would be great to assess further if machine learning models can be trained on lower-level data as peptides are the most sensible unit and imputation on protein groups level performs one form of implicit imputation at the peptide level[8]. One could assess if imputed features on lower-level data can be reaggregated to protein groups, e.g using ideas from Sticker and coworkers[45] or MSStats[46]. Additionally, the three self-supervised DL models could also be explored for denoising of samples, especially the generative VAE, or by adding diffusion models as they are trained by adding noise to the data[47].

Finally, an interesting application will be single cell proteomics with hundreds of MS runs. This community in proteomics has not yet developed their own methods to our knowledge, but might not want to fall back to the ones established for discrete count-based single cell RNA data[48,49] for intensity based label-free quantified proteomics data without further testing. In conclusion we suggest that holistic models such as the ones implemented in PIMMS can improve imputation for proteomics and that our evaluation workflow allows further experimentation leading to more robust imputation.

## Methods
### Description of the HeLa proteomics dataset
The HeLa cell lines were repeatedly measured as maintenance (MNT) and quality control (QC) of the mass spectrometers at Novo Nordisk Foundation Center for Protein Research (NNF CPR) and Max Planck Institute of Biochemistry. The samples were run as QC samples during the measurement of cohorts or as MNT samples after instrument cleaning and calibration using different column lengths and liquid chromatography methods. The cells were lysed by different protocols, which are expected to include digestion using trypsin, but on a per sample basis the exact protocol was not annotated[50]. The injection volume ranges from one to seven microliter.

Therefore, our dataset contains repeated measures of similar underlying biological samples acquired using DDA label-free quantification and can be used to explore general questions of applicability of self-supervised learning to proteomics data.

### Description of raw file processing of HeLa proteomics dataset
We used 564 raw files of quality and maintenance runs of HeLa cell lines from a larger set of 7444 quality control and maintenance runs. We processed all of these in a Snakemake[51] workflow as single runs in MaxQuant 1.6.12[24] yielding single abundances for precursor, aggregated peptide and protein group intensities using LFQ. As FASTA file the UNIPROT human reference proteome database 2019_05 release, containing 20,950 canonical and 75,468 additional sequences, was used for the DDA analysis. Contaminants were controlled using the default contaminants fasta shipped with Max-Quant. From the MaxQuant summary folder we then used the *evidence.txt* for precursor quantifications, *peptides.txt* for aggregated peptides and *proteinGroups.txt* for protein groups. The full dataset and detailed pre-processing steps are explained in a Data Descriptor[35].

### Feature selection strategy for quantified runs in general comparison workflow
We applied a two-step procedure for feature and sample selection (Supplementary Fig. 1). We used a cutoff of 25% feature prevalence across samples to be included into the workflow. Samples were then filtered in a second step by their completeness of the selected features. To be included a sample had to have 50% of the selected features. In

order to create train, validation and test splits, a dataset was split in the long-data view, where a row consists of a sample name, feature name and its quantification. We divided 90% of the data into training data, 5% into validation and 5% into the test split, including per default 75% MCAR and 25% MNAR simulated missing values. This ensured that the validation and test data were representative of the entire data, while enough low abundant intensities were available for evaluation of features from the lower range of intensities. The validation cohort was only used for early stopping and the performance on the validation and test data was therefore expected to be similar. On a few hundred sample datasets, the number of sampled quantifications for both validation and test split is quickly in the order of hundred thousand for protein groups and several hundred thousands for peptide-related measurements.

## GALA-ALD dataset

The clinical data consisted of a cohort of patients with liver disease[2]. 457 plasma samples were measured in data-independent acquisition (DIA) and processed using Spectronaut v.15.4[52] with the libraries as described in detail by Liu and coworkers[2]. Peptide quantification was extracted from "PEP.Quantity" - representing the stripped peptide sequence. Data for downstream analysis was selected with the same two-step procedure as described for the HeLa data. 3048 aggregated peptides were available in at least 25% of the samples of a total of 4345 aggregated peptides being present at least once. Protein group quantifications were extracted from "PG.Quantity", dropping filtered-out values. 377 protein groups were available in at least 25% of the samples of a total of 506 protein groups being present at least once. We used a fibrosis marker (kleiner[53] score ranging from zero to four, $N = 358$) to compare the effects of different imputation methods. In the original ALD study the features were further selected based on QC samples where a maximum coefficient of variation of 0.4 on the non log transformed quantification per feature was used as cutoff for inclusion. This step was omitted in the comparison with the original study results in Fig. 4 as we wanted to have a standardized workflow applicable also to approaches without interspersed QC samples. In numbers this means that we retained 313 protein groups instead of 277 omitting the selection criteria on QC samples. For the differential abundance analysis we had 348 complete clinical samples with both the kleiner score and the clinical control measurements we used.

## Self-supervised DL models

All models used self supervision as their setup, i.e. the data itself is used as a target in a prediction task. CF builds on the idea to combine a sample representation with a feature representation to a target value of interest[54–56]. The simplest implementation is to combine embedding vectors of equal length using their scalar product to the desired outcome, here the log intensity value assigned by a proteomics data analysis program. The approach is flexible to the total number of samples and features, and the model was trained only on the non-missing features. The loss function is the mean squared error.

A DAE is at inference time a plain autoencoder. During training its input values were partly masked and needed to be reconstructed. For each mini-batch the error was used to update the model so that the model learned better to reconstruct the data[57,58]. The loss was the squared error:

$$L_{reconstruction} = \sum_{i}^{N_B} \sum_{f}^{F_i} \left( I_{f,i}^{pred} - I_{f,i}^{obs} \right)^2 \tag{1}$$

where $N_B$ is the number of samples in a batch $B$, $F_i$ is the number of features not missing in a sample $i$ and $I_{f,i}$ is the predicted and observed label-free quantification intensity value $I$ of feature $f$ in sample $i$.

Missing features in a sample, which were not missing due to the training procedure of masking intensity values, were not used to calculate the loss. VAE introduces a different objective and models the latent space explicitly, here and as most often done as a standard normal distribution[59,60]. The latent space of a VAE has two components that are used for the first part of the loss function, the regularization loss:

$$L_{regularization} = \sum_{i}^{I_B} \sum_{l}^{L} \max\left(0, 0.5*\left\{\mu_{l,i}^z + e^{v_{l,i}^z} - 1 - v_{l,i}^z\right\}\right) \tag{2}$$

where $\mu_{l,i}^z$ is the mean and $v_{l,i}^z$ the log variance of dimension $l$ and sample $i$ of the isotropic multivariate Gaussian with $L$ dimensions of the encoder output, i.e. the latent representation z. The reconstruction loss was based assuming a normal distribution for the decoder as output[60,61], leading to

$$L_{reconstruction} = \sum_{i}^{N_B} \sum_{f}^{F_i} 0.5\left\{\ln(2\pi) + \left[\left(I_{f,i}^{obs} - \mu_{f,i}^I\right)^2\right]\cdot e^{-v_{f,i}^I} + v_{f,i}^I\right\} \tag{3}$$

where $N_B$, $F_i$ and $I_{f,i}^{obs}$ are as before and $\mu_{f,i}^I$ and log variance of $v_{f,i}^I$ are the parameters of the isotropic multivariate Gaussian distribution of the decoder outputs, i.e. of the modeled feature distribution. Training of the VAE was augmented by masking input values as in the denoising autoencoder[62], although this is not strictly necessary due to the stochastic nature of the latent space. For inference, missing values are predicted using both the mean of the encoder and decoder output. The models were developed using a variety of software including numpy (v.1.20)[63], pandas (v.1.4.)[64,65], pytorch (v.1.10)[66] and fastai (v.2.5)[55].

## Other imputation approaches

We used other methods which were available either in scikit-learn or R[14,67]. R or bioconductor packages were e1071, impute[36], SeqKnn[68], pcaMethods[69], norm, imputeLCMD[8], VIM[70], rrconNA[71], mice[72], missForest[38], GSimp[73] and msImpute[67] (Supplementary Table 1), which were included in a previous comparison[14] except the last one. We did not include non-general imputation methods as e.g. provided by MSstats[46] or without reusable software[74]. We used KNN interpolation of replicates based on the HeLa cell line measurements being repeated over time. The only parameter to set was how many neighboring samples should be used as replicates. We used three replicates for the scikit-learn[75] based implementation as this was found to be the best setting by Poulous and coauthors[26], which is also the most widely encountered replication number used in the field and set as default of both packages we used. However, the R based implementation of KNN (SEQKNN, TRKNN, KNNIMPUTE) used ten neighbors in NAguideR[14] and we kept the default. We also used a simple median calculation for each feature across samples. This requires estimating one parameter per feature. For features that did not vary a lot, this strategy should yield robust estimates for missing values. We used a random forest implementation using missForest[38]. The implementation works well for datasets on the protein group level, but fails for datasets on the peptide and precursor level as these are roughly ten times higher dimensional. We also included methods which assume MNAR missing values, such as the random shifted normal (RSN) distribution for imputation or QRLIC[21]. RSN has as parameters a global mean shift and scaling factor for standard deviation, as well as a mean and standard deviation for each unit of interest, i.e. all quantified features of a sample or for a feature all quantification of that feature across samples. Note, that RSN and other MNAR focused methods assume that measurements are not present as they are below the limit of detection (LOD). Therefore, in our default setup we sample 25 percent of the simulated missing values as MNAR into the validation and test data splits to represent the lower range of intensity values

(Supplementary Fig. 9), aiming to make the methods more comparable. All methods are available through our workflow and at least one method per included R package is tested via a GitHub action. Some methods and packages did however not work being called as in NAguideR for some of our data.

### Hyperparameter search using simulated missing values

In order to find good configurations for the self-supervised models a grid search was performed on three data levels on the development dataset. We sampled simulated missing values completely at random from the dataset, i.e. 5% for validation and 5% for testing. The training procedure and architecture of models was refined using the validation data. The performance of the best performing models on the validation data were then reported using the test data. We found that test performance metrics matched validation metrics up to the second decimal and that many model configurations yielded similar results. Performance was compared between the three self-supervised models to improve performance during model development. In this work all results were reported based on the simulated missing values in the test data split. Different latent representation dimension, namely 10, 25, 50, 75 and 100 dimensions were connected to a varying dimension and composition of hidden layers with a leaky rectified linear activation: (256), (512), (1024), (2056), (128, 64), (256, 128), (512, 256), (512, 512), (512, 256, 128), (1024, 512, 256), (128, 128, 128), (256, 256, 128, 128) - for the encoder and inverted for the decoder. The total number of parameters using these combinations ranged from a couple of ten-thousands in the case of the CF models to tens of millions for the autoencoder architectures. We picked the smallest model in terms of parameters of the top 3 performing ones as their performance was nearly equal on the validation data split. Then we retrained the best models with a share of 25 percent MNAR simulated missing values. Besides the best models on all simulated missing values, we reported results using other plots. The intensities in a split were binned by the median of the feature, e.g. protein group, they originated from based on the training data split. The MAE per bin was then reported (Fig. 2e, f), which allows for selection of the best models in the intensity range of interest. This is accompanied by a plot showing the proportion of missing values of a feature based on its median value over samples (Fig. 2c, d). The correlation plots were based on Pearson correlation of predicted intensities and their original values in the test split. The Pearson correlation was calculated for a feature across all predictions of all samples, denoted "per feature correlation", or for all predictions within one sample, denoted "per sample correlation" (Supplementary Fig. 6).

### Evaluation, imputation and differential expression in GALA-ALD dataset

We used the same splitting approach of the data as for the development dataset for evaluation with a share of 25 percent MNAR. We evaluated using a dimension of ten for CF's sample and feature embeddings, and the DAE and VAE latent spaces. The autoencoders were composed of one hidden layer with 64 neurons both for the encoder and decoder, leading to a total number of parameters between 9,174 and 74,462 for the three models. The RSN imputation for the missing values in the original ALD study was done on a per sample basis, i.e. mean and standard deviation for each sample. Using the two-step procedure with defaults as in the original study, this yielded 313 protein groups for comparison (see ALD data description). Using a filtering of 25% for feature prevalence prior to imputation with the VAE (Supplementary Fig. 1) we increased the share of missing values to 14% for the selected 377 protein groups in comparison to roughly 5% for the 313 features using the selection approach as in the original study[2]. The differential analysis was done using an analysis of covariance

(ANCOVA) procedure using statsmodels (v.0.12) and pingouin (v.0.5)[76,77]. We used a linear regression with the original kleiner score[53] as the stratification variable of interest for the patient's cirrhosis disease stage to predict protein quantifications, controlling for covariates. Therefore, effects for each protein group were based on an ANCOVA controlling for age, BMI, gender, steatosis, and abstinence from alcohol as well as correcting for multiple testing as done in the original study. The multiple comparison corrections (q-values) were based on 313 protein groups in the original data imputed using RSN, and on 377 protein groups retained here. Correction for multiple testing correction was done using Benjamini-Hochberg's correction[78]. The q-values of each DA were then compared for the overlapping 313 protein groups (Supplementary Data 10–12).

### Machine learning in GALA-ALD dataset

In order to assess the predictive performance of newly retained features, we evaluated a logistic regression using different feature sets for the binary target of a fibrosis score greater than one (F2 endpoint in the original study, False: kleiner <2, True, kleiner ≥2). The feature sets were: First, the features retained using the selection approach with settings as in the ALD study; second, all features available when using PIMMS selection approach; third, and the difference between both feature sets termed "new feat". We used maximum relevance, minimum redundancy using the F-test based implementation, in detail the F-test correlation quotient (FCQ)[42,79] to select a set of features to be used in the logistic regression. Using cross validation we selected the best set of up to 15 features for each of the three sub datasets. Then, the model was retrained on a final 80-20 percent training-testing data split of samples for each sub-dataset. Areas under the curve (AUC) for the receiver operation (ROC) and precision recall (PRC) curves were compared between these three sub datasets. The shown graphs and reported metrics were calculated on the test split[75,76].

### Reporting summary

Further information on research design is available in the Nature Portfolio Reporting Summary linked to this article.

## Data availability

The mass spectrometry proteomics data have been deposited to the ProteomeXchange Consortium via the PRIDE[80] partner repository with the dataset identifier PXD042233. A manuscript[50] describing these data has been published in Scientific Data and is available at https://doi.org/10.1038/s41597-024-02922-z. The clinical data is not freely available, but can be requested as specified by Niu et al.[2]: "The full proteomics datasets and histologic scoring generated and/or analyzed (…) are available (…) upon request, to Odense Patient Data Exploratory Network (open@rsyd.dk) with reference to project ID OP_040. Permission to access and analyze data can be obtained following approval from the Danish Data Protection Agency and the ethics committee for the Region of Southern Denmark." Source data for the main figures are provided with this paper. Source data are provided with this paper.

## Code availability

The PIMMS package and all analysis scripts are available on PyPI and at github.com/RasmussenLab/pimms[81]. The differential analysis and machine learning procedure used on the ALD data is available on PyPI and GitHub at github.com/RasmussenLab/njab.

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

## Acknowledgements

We would like to acknowledge Rebeca Quinones, Mario Oroshi and John Damm Sørensen for their huge effort to retrieve all quality and maintenance HeLa raw files for the creation of our development dataset. Furthermore, we would like to thank Jeppe Madsen and Martin Rykær for their time and expertise. Lastly, we would like to thank the proteomics groups that have kindly shared their maintenance and QC HeLa files, both at Novo Nordisk Foundation Center for Protein Research and at the Proteomics and Signal Transduction department of the MaxPlanck institute of biochemistry. H.W. was supported by the Novo Nordisk Foundation (grant NNF19SA0035440). H.W., A.B.N., L.N., M.L-P., M.M, L.J.J. and S.R. were supported by the Novo Nordisk Foundation grant NNF14CC0001 and S.R. from the Novo Nordisk Foundation grants NNF21SA0072102 and NNF23SA0084103.

## Author contributions

S.R. and A.B.N. initiated the study and guided the analysis. M.M. and A.B.N. led the efforts in collecting the data. H.W. assembled the data, performed the analyses and wrote the software choosing the modeling approaches and refining the idea. M.L.P, L.N. and L.J.J. provided guidance and input for the analysis. H.W. and S.R. wrote the manuscript with contributions from all coauthors. All authors read and approved the final version of the manuscript.

## Competing interests

The authors declare no competing interests.
