## [Peer Review File · Nature Communications]

Reviewers' Comments:

Reviewer #1:

Remarks to the Author:

This study proposes deep learning models for the imputation of the abundance of proteins, peptides, and precursors from mass spectrometry-based proteomics.

I have the following major concerns:

The title is not very clear and needs to be more specific. The term mass spectrometry-based proteomics refers to the whole research field, whereas this paper focuses on the imputation of the abundance of proteins, peptides, and precursors. Similarly, the phrases "proteomics imputation" or "imputation of proteomics data" may not clearly describe that the problem actually is.

Lines 104-119: the deep learning motivation is not well described. Here the authors provide generic descriptions of the deep learning models that they use for the imputation. From my perspective, I would be more interested in (1) the input data and the expected outputs from the models and (2) why deep learning models are suited for that kind of problem and data. The authors list some examples of previous deep learning applications for peptide feature prediction and tandem mass spectrum prediction. In those applications, the data format is amino acid sequences and they can be modelled using embedding and neural network models in deep learning. However, I do not see that connection in the protein abundance data and the authors did not discuss why that data could benefit from deep learning

Line 121: large (N=450) and small (N=50) datasets. The authors described two datasets but not mentioning their purpose, which is to evaluate the model performance on the sample size (this only becomes clear at a later section in the Results). From Figure 2a and 2c, I do not see significant difference of model performance between the two datasets, despite that their sizes are 10-fold different. Furthermore, I notice that this imputation problem seems to be a "small n, large p" problem, i.e. small sample size (~500 samples) and large feature space (5k proteins, 50k peptides/precursors). That makes me wondering whether deep learning is really a suitable approach to address this problem, because deep learning is well-known for its performance on large sample size (even the toy example of MNIST dataset already has 70k images). This brings us back to my previous question as why the protein abundance data could benefit from deep learning.

Lines 125-127: three deep learning models and two heuristic approaches were compared. The two heuristic approaches here are too naïve as they only consider the features independently and ignore their correlation. It is quite obvious that the abundance of proteins are highly correlated. Thus, the authors may need to include a baseline model that could take into account the correlation of proteins, i.e. a simple linear regression of a missing feature based on other observed features that are correlated with the missing one. Furthermore, the authors described the limitations of previous methods, e.g. the RSN, so it is better to include those methods into the comparison rather than using the heuristic approaches.

Reviewer #2:

Remarks to the Author:

The manuscript submitted by Weibel et al. proposes novel missing value imputation strategies for proteomics data. Missing values are a known issue and attempted to be solved by many publications in the past. To date, most methods only superficially address the problem mostly due to lack of knowledge about the systematics behind missing values as well as a lack of data following known distributions. As a result, heuristics are most commonly applied that only target systematic errors, leading to a known but at the moment accepted / acknowledged data quality loss for downstream analysis. Because of this, there is a great need for methods that tackle missing value imputation, particularly by avoiding utilizing heuristics. The paper is therefore an interesting contribution to this field and holds the potential to disrupt current practices. The abstract is clearly written and leads to the topic, providing an example usecase and a hypothesis that is about to be tested, including a conclusion drawn from the findings. The authors also provide

access to the source code via github, which is very clean and organized.

However, there are some issues which need to be addressed prior to a potential publication, particularly related to model training, evaluation, stability, robustness against initial condition, randomization and outliers.

Major issues:

- There is very few training data available for deep learning techniques in proteomics. It must be addressed how this impacts an (V)AE, since it is highly likely that it overfits on the training data. In general, no analysis is provided to assess the level of overfitting. Given that the used autoencoders either have a very large number of parameters, which could capture the full variability of the data, or very low number of parameters, which might solve the problem of overfitting to some extent, at the cost of modeling power, i.e. the model cannot capture the full extent of variability. I was not able to find such an analysis. In addition, a thorough error estimation / accuracy estimation for the AE/VAE is missing. Since there is no gold standard, the AE / VAE should be trained with permuted input data to estimate to which extent it overfits on noise only data. It should also be retrained (using different splits) multiple times to estimate the deviation in its loss and the impact of the initial conditions towards the imputed values, since these might dramatically differ if the model is too complex for too few training data. My suggestion is to retrain the model at least 10 times, then repeating the prediction, checking the standard deviation of individual imputed values and also check how the q-values of the downstream analysis are affected by this, or if different protein groups become significant. By doing so, one can show that the model is somewhat stable wrt to the initial conditions, randomization and that it potentially learns biological effects rather than finding a mathematically optimal solution.
- Simulated missingness is not necessarily representative of the real world distribution of missing values. In line 256, the authors describe how PIMMS was applied to real world data. I was not able to find a full description of how the model was trained, i.e. how the 455 blood plasma samples were split into train/val/test, training with masking the zeros and then evaluated only on the simulated missing values in the test split? Please provide a full description, making it very clear what was done. For the newly discovered differentially expressed genes, please provide swarm plots highlighting the imputed expression values including q-values to be able to assess whether the statistical significance is likely only due to the imputed values.
- AE/VAE are typically not capable of capturing the full scale of the input data as there is loss of information due to the low dimensional latent space. The question arises, what the scientific impact is, when the data is denoised (assuming the AE does what it is supposed to do) but does not capture the extent of variability in the individual data. I suggest added a thorough investigation of how much the model changes the measured expression values that should not be imputed.
- It is not clear to me how the training data was split and what data was used in the specific sections of the manuscript (train, validation, test). It is also not clear to me what exactly was used as input to the various AE models. I guess the input was a sample represented by a list of intensities of individual features? This is in part due to the sentence "On a few hundred sample data sets, the number of sampled quantifications for both validation and test split is quickly in the order of hundred thousand for protein groups and several hundred thousands for peptide related measurements (Supp. Table S2)." in methods. If the input to the AE is a vector of feature intensities, only very few samples are provided, that each have thousands of dimensions. This in turn leads to problems with regards to overfitting and not having a representative sample of the actual underlying biology. Table S2 does not clarify this.
- Figure 1b clearly shows additional batch effects, seen by the emerging structures within the main 3 areas. However, from the visualization is not apparent what this may be. I suggest generating multiple UMAPs with single coefficients being highlighted, e.g. time and instrument, separately. In addition, how does the UMAP change when applying imputation? Does the dataset show no batch effects anymore? If so, one may argue that relevant biology was learned. Otherwise, the model overfit to the batch effects visible. Given the difference in performance between the development and biological datasets, I would also expect to see a better performance of e.g. the median imputation on the development datasets after some batch effect correction methods was applied. This is further corroborated by the extensive filtering that was applied to select a subset of HeLa runs for training. Why do some proteins show a negative correlation across samples? How many of the QC runs trained on are actually good runs where not LC or MS problem was present during acquisition?

- As mentioned earlier, the level of overfitting of the model is not evaluated. As of right now, it is impossible to argue to which extent a trained model can be applied to new datasets and whether/if small changes in e.g. instrumentation or workflows would render a pre-trained model useless. The development dataset could be used to show to which extent a model can potentially be re-used. For example, train a model on one instrument, and apply it to all others. Train a model on a subset of samples acquired within a short range of time and apply to the rest.
 - Figure 3 shows that there is little benefit of using (V)AE over more simple methods, particularly median imputation. Where do the others see the particular advantage of the (V)AE then? How do differentially expressed proteins between VAE and median imputation vary? The others other compare VAE against RSN, arguably the worst method.
 - How did the authors make sure that no information leakage was present when training the VAE on the plasma proteome test? Following the argumentation that the VAE learns a low dimensional representation of the sample, the VAE should impute values that will follow this pattern. Training a model on the resulting differentially expressed proteins appears similar to the problem raised in <https://pubmed.ncbi.nlm.nih.gov/36004690/>.
 - In general, additional experimental validation is required to prove that the VAE is doing biologically correct imputation. Mathematical correctness itself does not imply biological meaning. A core problem here is that no differentiation between true and false zeroes is done, missing completely at random and missing not at random. The model is trained under the assumption that all features are present and thus must be imputed. This is not necessarily correct.
- Minor issues:
- The performance reported in Figure 1 appears to be biased by high intense features (MAE for VAE of 0.42 which is about the lowest observed for the rolling average). This is likely because missing values during training were sampled equally, thus they over represented features which are not missing (e.g. high intense).
 - line 52: I suggest expanding the description to the two categories, missing not at random (MNAR), e.g. LOD related missingness, and missing completely at random (MCAR) where we do not know any systematic reason.
 - line 63: This also depends on the instrument and the search engine used for peptide identification.
 - line 72: Suggest to use the term log-normal distributed.
 - line 86: Why do the authors argue that it is not possible to re-use models from scRNA data? scRNA-seq data is typically modeled using a negative binomial distribution computed as a poisson-gamma mixture, which is a continuous distribution. Similar to other distributions, a linker function can be applied (here log transformation) to get something closely related to a gaussian distribution. This principle is used in existing autoencoders for MVI in scRNA-seq data. This could also work for single cell proteomics, as long as there is enough data available.
 - line 127: I suggest rephrasing this as the 49 proteins found to be significantly differentially expressed may only be due to imputation, that doesn't mean that these are significant due to uncovered biology through the imputation. How many proteins would have been associated with fibrosis when randomly selecting proteins from the list of not differentially expressed ones?
 - In general, the figure labels and text are often very small. I suggest increasing the size for most of them.
 - Formulas for loss function: Please explain all individual characters and symbols in all formulas, e.g. "z".
 - In methods, the authors reference something with (ref. 24). Please remove the "ref.". Please also use the same font and font size throughout the manuscript.
 - I do not agree with the proposal that the data used for development will only be provided upon acceptance. Please upload and make the data available for the next revision.
 - The second to last paragraph in the Conclusion is very speculative and no evidence for most of the statements is provided in the manuscript. I suggest removing this.
 - Please replace the references to bioRxiv by the respective peer-reviewed publications.
 - Since best average compares very similar to the best model for each feature, I suggest only using the best average model to decrease complexity and reduce the redundant representation of 2e and 2f.

Reviewer #3:

Remarks to the Author:

This submission focuses on a timely and important subject: missing value imputation in label-free quantitative proteomics (LFQ). To tackle it, the authors propose an original path, based on deep learning. To avoid a loss of generalization power (owing to the diversity of proteomics data and to the amount of data necessary to train over-parametrized models), the authors leverage a self-supervised strategy (where the predictive model is trained in a supervised way using automatically generated labelled and masked data, as classically done in computer vision). In addition, the proposed strategy has many interesting features, like the capability to process large cohorts or to impute at different levels (ions, peptides, proteins).

Whilst I am positive about the approach, the current version of the article failed to convince me on three aspects:

- 1- The practical constraints of the approaches are not presented, evaluated or discussed.
- 2- There seems to have flaws in the evaluation setting.
- 3- The comparisons against pre-existing methods are not sufficient.

These elements, detailed below, must be addressed before considering a publication.

Details:

1- The authors must be more explicit about the applicability of their approaches, notably for an interdisciplinary venue like Nature Communications. In the current state, the reader has no information about the computational load and the execution time (including the self-training step) and how these scale depending on the data size (numbers of features, samples, MVs, etc.). Similarly, most of LFQ experiments relies on a handful of samples within each biological group (3 to 10), making the imputation both difficult and necessary. Conversely, the authors evaluate their methods on datasets with 50 to 450 samples, for which the imputation can be expected to be easier and not as necessary. Thus, it is currently impossible to assess whether it makes sense to apply one of the proposed methods to a "classical" LFQ dataset. As any method, this one must have limitations, and it is not inherently a problem. However, it is important to clearly formulate these limitations (and to support them experimentally), as they will dramatically influence the concrete interest of Nat Comm readership.

2- The simulation experiments (to estimate the MAE) are not sufficiently described. From my understanding, the masked MVs have been randomly chosen as a result of a classical train/validation/test split. However, doing so results in only generating MCAR values for subsequent evaluation. MCAR values can be efficiently imputed using pre-existing methods (e.g. random forest -- <https://www.nature.com/articles/s41598-021-81279-4>), but on the other hand, they are not realistic (LOD related MVs are not MCARs). To fairly assess the performances of an imputation method, it is necessary to compare the MAE on masked datasets with a varying ratio of MCAR/MAR (like in refs. 13 or 17).

Likewise, the ALD experiment, though interesting, is hardly convincing: the proposed approach allows to discover more differentially abundant proteins, including a handful of biologically relevant ones, but the discussion does not account for the false positives. On the other hand, when this information is accounted for (like in fig 3e) the interest of the proposed approach seems marginal, notably at low FPR.

3- The anterior literature is not adequately accounted for. Although all the relevant articles are referenced, they are not correctly cited/contextualized. Notably, according to the authors, proteomics imputation is essentially conducted at protein level with the assumption that missing values result from a detection limit (like with the default Perseus pipeline), but the conclusions of refs 12-18 are more nuanced (to put it mildly) and these should have been considered in the experimental evaluations. Concretely, the authors compare their 3 methods against 3 baseline methods: median imputation (a notoriously poorly performing method), interpolation (which is not an imputation method per se, as a time series structure must be assumed, contrarily to most of the other methods) and RSN (not applied to the HeLa dataset). However, a host of alternative (and more refine) approaches have been published, each of them performing differently depending on the amount of MVs, their MCAR/MAR nature, the imputation level (ion, peptide or protein), etc. For instance, the NAGuideR paper (ref 18) lists 15 to 20 methods and proposes a unified framework to compare them. It is of the utmost necessity the authors comprehensively evaluate on both

datasets (HeLa and ALD) their methods against all the state-of-the-art methods, not only a couple of notoriously poor performing ones.

To summarize, the authors proposed a new appealing approach to LFQ imputation, but in the current state of the manuscript, the presented experimental evaluations are not sufficient to demonstrate its practical applicability (point 1), its accuracy in various realistic MV settings (point 2) and how it performs with respect to pre-existing methods (point 3).

Thanks a lot for taking the time to review our article. It's an investment of your time which we highly appreciate. The comments were all formulated fairly and we especially appreciate the explicit suggestions for improvements for each concern that was raised.

In the light of your comments, we want to emphasize here that our models are not trained once and then used as a general purpose model (in comparison to e.g. large language models which got a lot of attention the last months). Given a dataset, a model with a specified architecture is trained to learn the data, which can take 15 seconds up to 10 minutes depending on many parameters, including loading data and saving the imputations.

Major changes to the manuscript:

1. We added most imputation methods listed in NAGuideR to our comparison workflow, which now also can be used for imputation using PIMMS.
2. We changed the figure for inspecting the performance across the dynamic range of the features.
3. We investigated training on permuted data (per feature) and how stable the performance is on different splits and/or repeated model fits.
4. We added additional comparisons on smaller HeLa datasets (N=10, N=20, N=30, N=40).
5. Intensities were removed at random from the ALD data and the possibility to recover such signal loss by models for the downstream analysis task was investigated.
6. Swarm plots of non-imputed and imputed intensities for the ALD data were added, and a selection included into the paper.

Reviewer #1 (Remarks to the Author)

This study proposes deep learning models for the imputation of the abundance of proteins, peptides, and precursors from mass spectrometry-based proteomics.

I have the following major concerns:

The title is not very clear and needs to be more specific. The term mass spectrometry-based proteomics refers to the whole research field, whereas this paper focuses on the imputation of the abundance of proteins, peptides, and precursors. Similarly, the phrases "proteomics imputation" or "imputation of proteomics data" may not clearly describe what the problem actually is.

The title was chosen broadly as the discussed methods are in general applicable to any matrix of samples (rows) with the corresponding features (columns). As we however only test their application on the three mentioned data levels, we decided to change the title to "Imputation of label-free quantitative mass spectrometry-based proteomics data using self supervised deep learning"

Lines 104-119: the deep learning motivation is not well described. Here the authors provide generic descriptions of the deep learning models that they use for the imputation. From my perspective, I would be more interested in (1) the input data and the expected outputs from the models and (2) why deep learning models are suited for that kind of problem and data. The authors list some examples of previous deep learning applications for peptide feature prediction and tandem mass spectrum prediction. In those applications, the data format is amino acid sequences and they can be modelled using embedding and neural network models in deep learning. However, I do not see that connection in the protein abundance data and the authors did not discuss why that data could benefit from deep learning

We adapted the section in order to describe the expected input and output of models. If the models are suited for imputation is what we try to investigate in this study. We think that the models are able to create representations of the underlying data, which are indeed intensities on the precursor, peptide or protein groups level. The representations are either separate feature and sample embeddings or a joint embedding space and allow to holistically model the data. The generated representations of the data - the embeddings - could then be used in further analysis. We therefore argue that the holistic models can, in the absence of a clear statistical model of the data generating process, be leveraged for tasks w.r.t the data distribution. Deep learning models are believed to have a high complexity, even allowing to fit random labels to a certain extent. The implication is that assessing the model's generalization capabilities is hard and that only good data should be used. In principle, this will also apply to any other framework, which is why agreeing on an evaluation strategy and providing a framework to do so is important in our opinion. In comparison to the other mentioned deep learning models we do not work on amino acid sequences representing each amino acid by a feature vector, but rather represent an amino acid sequence in its entirety as a key in a feature vector of intensities.

This part of the introduction now reads:

Lines 102-109: "We turn to machine learning for imputation of missing intensities as it offers the possibility to learn from the data itself. Deep Learning (DL) has been used to improve over existing machine learning models in a variety of biological data problems²⁷⁻²⁹. DL has been successfully applied to predict peptide features such as retention times, collisional cross-sections and tandem mass spectra, significantly boosting the peptide identifications and precision of searches of MS-based proteomics³⁰⁻³⁴. We use intensities from search results generated from tools like e.g. MaxQuant and Spectronaut and apply DL methods to impute this MS-based proteomics data. We considered three types of models that process the search results of precursors, peptides or protein groups slightly differently."

Line 121: large (N=450) and small (N=50) datasets. The authors described two datasets but not mentioning their purpose, which is to evaluate the model performance on the sample size (this only becomes clear at a later section in the Results). From Figure 2a and 2c, I do not see significant difference of model performance between the two datasets, despite that their sizes are 10-fold different. Furthermore, I notice that this imputation problem seems to be a "small n, large p" problem, i.e. small sample size (~500 samples) and large feature space (5k proteins, 50k peptides/precursors). That makes

me wondering whether deep learning is really a suitable approach to address this problem, because deep learning is well-known for its performance on large sample size (even the toy example of MNIST dataset already has 70k images). This brings us back to my previous question as why the protein abundance data could benefit from deep learning.

We apologise for not mentioning that the goal of using the two HeLa datasets was to test the dependence on sample size. We now include that in the first mentioning of the datasets. Additionally we added N=10, N=20, N=30 and N=40 samples comparison with the setup of our models as for the N=50 data (Supp. Data. 3.)

That the performance of the models on the datasets (N=50 and N=500) are the same indicates that a N of 50 is sufficient to train the models for the HeLa dataset. With regard to the “small n, large p” problem, we, interestingly, see that the higher dimensional data (peptides/precursors) yield better results (lower MAEs) compared to the protein group data. While some of this can be attributed to the aggregation step in creating protein groups (or peptides) this also points towards that the sample sizes are high enough to generalize within the datasets. Additionally, we use data augmentation on the training data to increase the variability of the data and further regularize the models by using dropout. This is important as we do not want to generalize to other datasets from these data but rather retrain the model on each dataset that it is applied to. This is an approach that we have had good experience with for small N, large p datasets (Nissen et al., Nature Biotechnology 2021 and Allesøe et al., Nature Biotechnology 2023).

We have updated the text with the following text:

Lines 202-214: “Reducing the training sample by roughly a tenth, performance on the smaller development dataset (N=50) was comparable to the performance on the larger development dataset, while runtime reduced by 5 to 15 times for precursors (Median: 0:23, KNN: 0:33, VAE: 1:04, DAE: 0:52, CF: 1:06). For the smaller development dataset on the protein groups level, additional R methods from NAGuideR were able to complete. We found that BPCA (Oba et al. 2003) performed best on the 50 homogenous HeLa samples with relatively long runtime (e.g. on precursors: BPAC: 20:44, MICE-NORM: 4:42). For 10, 20, 30 and 40 samples, other methods showed good results, although PIMMS models still fit the data (**Supp. Data 3**): For example with 10 samples the precursors were fitted best with an MAE of 0.57 using BPCA and 0.61 using CF. In summary, this indicated that on an unnormalized, intensity varying dataset from a single machine the PIMMS models were able to capture patterns between detected features to impute values (**Supp. Fig. S3a**) for as few as 50 samples with competitive performance to other state-of-the-art methods.”

Lines 125-127: three deep learning models and two heuristic approaches were compared. The two heuristic approaches here are too naive as they only consider the features independently and ignore their correlation. It is quite obvious that the abundance of proteins are highly correlated. Thus, the authors may need to include a baseline model that could take into account the correlation of proteins, i.e. a simple linear regression of a missing feature based on other observed features that are correlated with the missing one. Furthermore, the authors described the limitations of previous methods, e.g. the

RSN, so it is better to include those methods into the comparison rather than using the heuristic approaches.

This is a good point and we have now included several other methods in the comparisons (Supp. Table 1). For example, we used the scikit-learn KNN- imputation procedure that allows for having different nearest neighbors for a single sample depending on which feature is imputed (potential “donors” are defined per feature), an SVD imputation which is a linear approximation of the data using functionality referenced in NAGuideR, and a random forest from the proteomics imputation referenced in software NAGuideR (Wang et al. 2020). All NAGuideR methods which could be installed, were added to PIMMS. However, we did encounter that some of the methods failed, did not converge or had too long runtimes on larger datasets (Supp. Data 3,4). We also added comparisons with N=10, N=20, N=30 and N=40 for the three data levels on the HeLa data. With regards to RSN we did not include it originally in the analysis of the HeLa datasets as it is focused on missing not at random (MNAR). Therefore, the assumptions of RSN are violated for that dataset and with the current evaluation setup of HeLa RSN performs by design 4-5 times worse as all non-low-abundant features will be imputed wrongly.

Therefore, we now compare the three DL methods to a wide range of methods, although not all are suited for MAR missing values (Supp. Data 3). We have updated Figures 2 and 4 (former 3), supplementary figures 4, 5, 7 and 8 and modified the following in the text mentioned in the introduction:

Lines 126-127: “For evaluation, we developed a workflow that allowed comparison between the three DL models and other approaches listed in the NAGuideR tool (Supp. Table 1).”

Reviewer #2 (Remarks to the Author)

The manuscript submitted by Weibel et al. proposes novel missing value imputation strategies for proteomics data. Missing values are a known issue and attempted to be solved by many publications in the past. To date, most methods only superficially address the problem mostly due to lack of knowledge about the systematics behind missing values as well as a lack of data following known distributions. As a result, heuristics are most commonly applied that only target systematic errors, leading to a known but at the moment accepted / acknowledged data quality loss for downstream analysis. Because of this, there is a great need for methods that tackle missing value imputation, particularly by avoiding utilizing heuristics. The paper is therefore an interesting contribution to this field and holds the potential to disrupt current practices. The abstract is clearly written and leads to the topic, providing an example use case and a hypothesis that is about to be tested, including a conclusion drawn from the findings. The authors also provide access to the source code via github, which is very clean and organized. However, there are some issues which need to be addressed prior to a potential publication, particularly related to model training, evaluation, stability, robustness against initial condition, randomization and outliers.

Major issues:

There is very few training data available for deep learning techniques in proteomics. It must be addressed how this impacts an (V)AE, since it is highly likely that it overfits on the training data. In general, no analysis is provided to assess the level of overfitting. Given that the used autoencoders either have a very large number of parameters, which could capture the full variability of the data, or very low number of parameters, which might solve the problem of overfitting to some extent, at the cost of modeling power, i.e. the model cannot capture the full extent of variability. I was not able to find such an analysis. In addition, a thorough error estimation / accuracy estimation for the AE/VAE is missing. Since there is no gold standard, the AE / VAE should be trained with permuted input data to estimate to which extent it overfits on noise only data. It should also be retrained (using different splits) multiple times to estimate the deviation in its loss and the impact of the initial conditions towards the imputed values, since these might dramatically differ if the model is too complex for too few training data. My suggestion is to retrain the model at least 10 times, then repeating the prediction, checking the standard deviation of individual imputed values and also check how the q-values of the downstream analysis are affected by this, or if different protein groups become significant. By doing so, one can show that the model is somewhat stable wrt to the initial conditions, randomization and that it potentially learns biological effects rather than finding a mathematically optimal solution.

For the issue of overfitting we handle it using the following approach: First, we do not intend to transfer the model to other data, i.e. we train the model on the dataset that we want to impute on. If one wants to impute another dataset one needs to retrain the model from scratch on that dataset and then do the imputation. This is a similar approach to what we have done for microbiome and multi-omics datasets (Nissen et al., Nature Biotechnology 2021 and Allesøe et al., Nature Biotechnology 2023). Second, to prevent overfitting we have a validation and training data split. The validation data is used to stop learning if no more progress is made on the validation data while training on the training data. The test data should then give a rough estimate of model performance and is performed for each new dataset the models are fitted on. In general in our experience developing PIMMS and the other methodologies it is hard for the VAE to overfit. This is in agreement with Feldman and Chang which shows that general over parameterized, but regularized deep learning models, are believed to perform well (Feldman and Zhang 2020) .

We have now added the following to the main text:

Lines 172-179: “We ran several configurations using a grid search to find the best configurations using simulated missing values in a validation and test split from all samples (see Methods, **Supp. Data 1, 2**). In absolute numbers these were 100,001 for the protein group, 616,561 aggregated peptides and 661,817 precursors intensities in the test set for our development dataset of instrument 6070. When investigating the performance of the imputation methods we used the mean absolute error (MAE) on the log₂ scaled intensities between predicted and true measured intensity values on our simulated missing values.”

Following the suggestion on testing the overfitting capabilities, we created a dataset of permuted features by randomly shuffling protein groups one by one on the large HeLa development dataset. This means that the median performance does not change. In **Supp. Fig. S6a**, we show that on this data the median is indeed better than using the models for imputation indicating that the machine learning models are not able to learn from randomized data. Additionally, we retrained the CF, DAE and VAE five times for the non-shuffled large HeLa development dataset both on the same split and on a new random split to show the training variance (**Supp. Fig. S6c,b**). The results show that retraining on the same data split or newly splitted data yields nearly identical performances. See also **Supp. Data 6**.

The standard deviation for imputed values (here visualized in terms of the coefficient of variation is rather low for all methods (here for the missing values imputes 10x in the ALD dataset)

We added the following to the main text:

Lines 235-244: “Finally, we investigated how stable the self-supervised models trained. In a first step, we randomly permuted all protein groups of the large development dataset, which leaves the median performance unchanged. Accordingly, training models on randomly permuted data could not outperform the imputation by the median of the training data split with an MAE of 1.04 (**Supp. Fig. S5a, Supp. Data 5**, CF: 1.05, RF, VAE: 1.08, DAE: 1.12). Next, we trained the self-supervised models five times on the same data split as well repeated training five times on new splits. The performance between fitted models varied in a narrow margin (CF: 0.427-0.433, DAE: 0.424-0.436, VAE: 0.438-0.453 [Min-Max]) (**Supp. Fig. S5b,c, Supp. Data 5**). We therefore conclude that the DL models could be consistently fitted to data and that self-supervised models were able to fit the data holistically for imputation purposes.”

With regard to replication of q-values we added an additional analysis where we removed 20 percent of the data assuming MAR. Briefly, the self-supervised models were able to recover lost signal due to MAR missing values. For more details see response to reviewer #3 below.

Simulated missingness is not necessarily representative of the real world distribution of missing values. In line 256, the authors describe how PIMMS was applied to real world data. I was not able to find a full description of how the model was trained, i.e. how the 455 blood plasma samples were split into train/val/test, training with masking the zeros and then evaluated only on the simulated missing values in the test split? Please provide a full description, making it very clear what was done. For the newly discovered differentially expressed genes, please provide swarm plots highlighting the imputed expression values including q-values to be able to assess whether the statistical significance is likely only due to the imputed values.

We agree that this should be clarified. For the real world dataset (ALD) we followed a similar setup as with the HeLa data. We used the Spectronout protein group intensities, applied a log two transformation and then sampled from all intensities in the table (i.e. across all samples) training, validation and test data. We trained the models on the training data using early stopping on the validation data. Results are then reported on the test data (and the model architecture is indicated on the bar plots).

Across all samples and protein groups we selected 90% of intensities as training data, 5% as validation and 5% as test data. This corresponds to 133,116, 7398, 7399 intensities, respectively. For the differential analysis we imputed missing values in all samples based on the models trained from the training split. We now thoroughly explain this in the Methods section:

Lines 661-668: “We sampled simulated missing values from the dataset, i.e. 5% for validation and 5% for testing. The training procedure and architecture of models was refined using the validation data. The performance of the best performing models on the validation data were then reported using the test data. We found that test performance metrics matched validation metrics up to the second decimal and that many model configurations yielded similar results. Performance was compared between the three self-supervised models to improve performance during model development. In this work all results were reported based on the simulated missing values in the test data split.”

Finally, we created for the differentially identified abundant protein groups swarmplots of the observed and imputed abundances. These are shown in Figure 4d-g for four protein groups (F5H8B0;P08709;P08709-2, A0A0D9SG88, P06702 and Q8WUD1;Q8WUD1-2) where the differential abundant decision differed between the imputation methods. It highlights that RSN per sample distorts some decision by an out of distribution shift, and that for newly included protein groups we can observe gains by more data driven models.

In the main text we added:

Lines 379-385: “Few relatively non-prevalent protein groups, i.e. here below 70% prevalence (250/358 and less) of quantified samples, showed a strong difference in differential analysis testing between e.g. the VAE and the RSN imputation. The relatively non-prevalent (222/358) protein group F5H8B0/P08709/P08709-2 associated with the gene *F7* was clearly not significant using the VAE, but passed the statistical thresholds when using RSN imputation. On the other hand, a relatively rarely

quantified protein A0A0D9SG88 (gene *CFH*) was significant using VAE imputation, but not RSN (**Supp. Data 10**). Visualization using swarm plots indicated that the RSN imputation for these two protein groups were shifted downwards of the original center (**Fig. 4d-e**). In combination with a slight imbalance of targets in the 142 imputed samples, this led to the different results. Similarly, for two newly included protein groups P06702 (gene *S100A9*) and Q8WUD1/Q8WUD1-2 (gene *RAB2B*) imputation by a fixed median value did not lead to significant results (**Fig. 4f-g, Supp. Data 10**)."

AE/VAE are typically not capable of capturing the full scale of the input data as there is loss of information due to the low dimensional latent space. The question arises, what the scientific impact is, when the data is denoised (assuming the AE does what it is supposed to do) but does not capture the extent of variability in the individual data. I suggest added a thorough investigation of how much the model changes the measured expression values that should not be imputed.

This is an important point and we agree that it would require more investigation. We do however not change any of the original values, but only use the imputed missing values (see Supp. Fig. 8 for an overall distributional comparison). Therefore, no original intensity value was changed for the downstream analysis and denoising of the data is not done as we only focus on imputation. To clarify this we have now added the following to the main text:

Lines 342-346: "By using these models for imputation of missing data, leaving the available data as it was, we were able to perform the analysis for 377 protein groups (ANCOVA, Benjamini-Hochberg multiple testing correction, $p\text{-value} \leq 0.05$) compared to only 313 protein groups when using the RSN imputation approach as originally applied in Niu et. al (2022)²."

It is not clear to me how the training data was split and what data was used in the specific sections of the manuscript (train, validation, test). It is also not clear to me what exactly was used as input to the various AE models. I guess the input was a sample represented by a list of intensities of individual features? This is in part due to the sentence "On a few hundred sample data sets, the number of sampled quantifications for both validation and test split is quickly in the order of hundred thousand for protein groups and several hundred thousands for peptide related measurements (Supp. Table S2)." in methods. If the input to the AE is a vector of feature intensities, only very few samples are provided, that each have thousands of dimensions. This in turn leads to problems with regards to overfitting and not having a representative sample of the actual underlying biology. Table S2 does not clarify this.

We apologize for the confusion. The input to the autoencoder based models (DAE, VAE) are indeed matrices with samples in the rows and the features in the columns. This is what we term "wide" data format. In contrast, the CF model uses the long data format, where pairs of samples and features are input with their recorded intensity.

To clarify on the train, validation and test data split we used 90% training data split, a 5% validation data split and a 5% testing data split. Here each observation is an intensity of a pair of a sample and a feature. We sample from the overall distribution of features pairs of samples and features in a test and validation split which have the same distribution as the original data. We added a new supplementary figure (**Supp. Fig. S9**) that shows the distributions of the training, validation and test data splits in the small HeLa dataset (4,405 protein groups of N=50). Additionally **Fig. 4a** shows the splits for the full ALD dataset

Fig. S9: Stacked histogram of train, validation and test split for protein groups of small development dataset. a) Histogram of validation and test data split. b) training data distribution versus test data distribution c) Stack histogram of all three splits. Bins were created per integer of log₂ transformed intensities. Number of intensities per split in parentheses.

Additionally, we now tried to clarify this in the methods section:

Lines 661-668: “We sampled simulated missing values from the dataset, i.e. 5% for validation and 5% for testing. The training procedure and architecture of models was refined using the validation data. The performance of the best performing models on the validation data were then reported using the test data. We found that test performance metrics matched validation metrics up to the second decimal and that many model configurations yielded similar results. Performance was compared between the three self-supervised models to improve performance during model development. In this work all results were reported based on the simulated missing values in the test data split.”

Figure 1b clearly shows additional batch effects, seen by the emerging structures within the main 3 areas. However, from the visualization is not apparent what this may be. I suggest generating multiple UMAPs with single coefficients being highlighted, e.g. time and instrument, separately. In addition, how does the UMAP change when applying imputation? Does the dataset show no batch effects anymore? If

so, one may argue that relevant biology was learned. Otherwise, the model overfit to the batch effects visible. Given the difference in performance between the development and biological datasets, I would also expect to see a better performance of e.g. the median imputation on the development datasets after some batch effect correction methods was applied. This is further corroborated by the extensive filtering that was applied to select a subset of HeLa runs for training. Why do some proteins show a negative correlation across samples? How many of the QC runs trained on are actually good runs where no LC or MS problem was present during acquisition?

The original figure was showing data from multiple machines, however we only used data from one machine, instrument 6070. We therefore replaced Fig. 1b and 1c with PCAs of the development dataset originating from that machine (instrument 6070). Applying imputation to a dataset will not remove batch effects as the entire data distribution is learned. Outliers will be regressed towards the mean but this does not imply an entire repositioning of samples. If we would test for denoising all intensities would need to be replaced by the predicted intensities, i.e. the data used for learning will be changed. For now we only focused on imputing missing values.

Regarding negative correlations: Some simulated missing values for a protein group can have negative correlations if they e.g. fall below the mean of the predicted intensities and above the mean for the observed intensities (or vice-versa).

For the quality of QC runs, we used overall sample completeness of the selected features as a definition of quality. LC or MS problems are not part of the metadata, but have to be inferred indirectly. The models will depend on the quality of the dataset. In the real world application the quality of the samples was checked in contrast thoroughly and their measurements are better aligned (**Supp. Fig. S3**).

We clarified in the main text:

Lines 211-214: "In summary, this indicated that on an unnormalized, intensity varying dataset from a single machine the PIMMS models were able to capture patterns between detected features to impute values (**Supp. Fig. S3a**) for as few as 50 samples with competitive performance to other state-of-the-art methods."

As mentioned earlier, the level of overfitting of the model is not evaluated. As of right now, it is impossible to argue to which extent a trained model can be applied to new datasets and whether/if small changes in e.g. instrumentation or workflows would render a pre-trained model useless. The development dataset could be used to show to which extent a model can potentially be re-used. For example, train a model on one instrument, and apply it to all others. Train a model on a subset of samples acquired within a short range of time and apply to the rest.

As mentioned above, we retrain a model for each dataset, i.e. having dataset specific models. This worked well elsewhere (Nissen et al. 2021; Allesøe et al. 2023) and is in line with other imputation techniques where local or global statistics are computed on the dataset at hand for imputation. That

said, there could be a setting where extrapolation of a model could make sense, but as correctly stated, then the assessment on entirely new samples should be added.

We envisage that future work could explore how one could train on an extensive Proteomics database with data generated from comparable FASTA files in a similar fashion to RNA-seq based foundation models.

We now clarified this:

Lines 506-515: “In general, the modeling approaches here are restricted to the samples in a particular study and all models are fitted for each new dataset. However, transfer of models between datasets can be envisioned. The potential to fine-tune a model trained on one dataset to a new dataset for a fixed set of features is possible without further efforts for autoencoders. For CF it would need to find the closest training samples in the case where samples are separated strictly into train, validation and test set. However feature embeddings could be transferred and extended easily. Therefore, all models could potentially be envisioned in a clinical setup, where models are re-trained with the latest samples. This could be implemented by the use of similar cohorts, e.g. for the same tissue and similar patients, which is then the basis to build a database of tissue specific models - or by incorporating tissue embeddings as an additional source of information.”

Figure 3 shows that there is little benefit of using (V)AE over more simple methods, particularly median imputation. Where do the authors see the particular advantage of the (V)AE then? How do differentially expressed proteins between VAE and medium imputation vary? The authors compare VAE against RSN, arguably the worst method.

We initially selected RSN both because it is commonly used (also in the original study) and we saw that the imputations for missing values were down shifted, also for the VAE model (**Supp. Fig. S7**). We added a random forest, a k-nearest neighbor and the median to the comparison. We see that the CF and (V)AE adds more conditioned predictions than the constant median imputation (which would be the same after randomly shuffling the intensities of a protein group), thus leading to a better analysis. We therefore see the main advantage of the (V)AE to be able to learn to connect all necessary features at once.

Additionally, relying only on median imputation could mean that potential patterns are missed in a differential analysis:

Lines 388-390: “Similarly, for two newly included protein groups P06702 (gene *S100A9*) and Q8WUD1/Q8WUD1-2 (gene *RAB2B*) imputation by a fixed median value did not lead to significant results (**Fig. 4f-g, Supp. Data 10**).”

As mentioned above, we added an analysis where we remove 20% of the data at MAR. We showed that CF, DAE and VAE could recover more differentially abundant protein groups than the simple median (**Supp. Fig. S5**). See Fig. 3 and Supp. Data 8 for a full overview and the results from repeating the differential analysis. The results are described in the main text.

Lines 308-325: “Then we tested the ability of the methods to recover differentially abundant protein groups when removing data at random (MAR). Therefore, we randomly removed 20 percent of the data. Comparing the results of the differential abundance analysis using the complete dataset compared to the reduced dataset we observed a decrease in the number of differentially abundant protein groups from 212 to 192 (ANCOVA, Benjamini-Hochberg multiple testing correction, $p\text{-value} \leq 0.05$). Interestingly, we found that besides 22 protein groups losing their differential abundance, four became differentially abundant by randomly removing data. To assess the effectiveness of various imputation methods, we used no imputation as a benchmark. We found that DAE, VAE, and RF imputation techniques successfully recovered 20 out of the 22 protein groups that lost their differential abundance. In contrast, median imputation only recovered two protein groups, while CF imputation recovered 16 groups with three protein groups slightly above the cutoff of 0.05. KNN recovered 13 protein groups (**Fig. 3c,e, Supp. Data 8**). Four protein groups became differentially abundant without imputation when intensities were removed in the 80% dataset, which we labeled as false positives to follow our approach (**Fig. 3d,f**). However, three of the four were nearly statistically significantly abundant in the 100% dataset and therefore adding back patients through imputation to the analysis can be seen as beneficial. This indicated that the PIMMS models were capable of recovering lost signals of at least MAR data.”

How did the authors make sure that no information leakage was present when training the VAE on the plasma proteome test? Following the argumentation that the VAE learns a low dimensional representation of the sample, the VAE should impute values that will follow this pattern. Training a model on the resulting differentially expressed proteins appears similar to the problem raised in <https://pubmed.ncbi.nlm.nih.gov/36004690/>.

As mentioned above, no transfer learning was intended as this would mean among other reasons that everyone needs to have the same proteome and it would assume that all features should be imputed based on a reference proteome.

We do not see the risk of information leakage as we only imputed missing values and left all original values as they were. The patterns in the data guide the training of self supervised models and one would expect to see imputations based on the patterns in the data (see Supp. Figure S8).

The mentioned article is a great warning. We however note that the article describes a one vs all cross validation in small sample size (~40). It also mentions that the more classic X-fold split variant of cross validation successfully prevents the undesired behavior: Our ALD prediction models were evaluated using 5 fold, 10 times cross validation before a final model was trained. Features were selected only on

the training data in each iteration (training data folds). To all our knowledge the selected features therefore need to have a degree of generalization to predict the target of interest.

We clarify this in the main text:

Lines 442-447: “Using minimum redundancy, maximum relevance (MRMR) approach we selected the most predictive set of features of each subset of features on the assigned training samples³⁹. Using all available protein groups median imputation yielded the best area-under the receiver operating curve (AUROC) of 0.90 compared to 0.87-0.90 for DAE, VAE and CF on the test samples (**Supp. Fig. S8**).”

In general, additional experimental validation is required to prove that the VAE is doing biologically correct imputation. Mathematical correctness itself does not imply biological meaning. A core problem here is that no differentiation between true and false zeroes is done, missing completely at random and missing not at random. The model is trained under the assumption that all features are present and thus must be imputed. This is not necessarily correct.

As mentioned above, in order to test the biological correctness, we added an additional experiment on the ALD dataset. We randomly removed 20 percent of the data MAR. We conclude that lost signal can be recaptured using the CF, DAE and VAE.

Lines 308-325: “Then we tested the ability of the methods to recover differentially abundant protein groups when removing data at random (MAR). Therefore, we randomly removed 20 percent of the data. Comparing the results of the differential abundance analysis using the complete dataset compared to the reduced dataset we observed a decrease in the number of differentially abundant protein groups from 212 to 192 (ANCOVA, Benjamini-Hochberg multiple testing correction, $p\text{-value} \leq 0.05$). Interestingly, we found that besides 22 protein groups losing their differential abundance, four became differentially abundant by randomly removing data. To assess the effectiveness of various imputation methods, we used no imputation as a benchmark. We found that DAE, VAE, and RF imputation techniques successfully recovered 20 out of the 22 protein groups that lost their differential abundance. In contrast, median imputation only recovered two protein groups, while CF imputation recovered 16 groups with three protein groups slightly above the cutoff of 0.05. KNN recovered 13 protein groups (**Fig. 3c,e, Supp. Data 8**). Four protein groups became differentially abundant without imputation when intensities were removed in the 80% dataset, which we labeled as false positives to follow our approach (**Fig. 3d,f**). However, three of the four were nearly statistically significantly abundant in the 100% dataset and therefore adding back patients through imputation to the analysis can be seen as beneficial. This indicated that the PIMMS models were capable of recovering lost signals of at least MAR data.”

Minor issues:

The performance reported in Figure 1 appears to be biased by high intense features (MAE for VAE of 0.42 which is about the lowest observed for the rolling average). This is likely because missing values during

training were sampled equally, thus they over represented features which are not missing (e.g. high intensity).

Yes that is true, which is why we initially added the rolling average error plot. We replaced it by an error plot showing binned intensity values in the test data split medians of a feature (**Fig. 2e-f, 4c**). The performance is indeed better for the bulk of the data and rare features are performing worse. Partly, the MAE's uncertainty increases for intensities from low and highly abundant features due to reduced number of intensities in these bins. We want to highlight that model evaluation using simulated missing values is aiming at learning the overall distribution.

Lines 216-224: "We evaluated the performance across the dynamic range of intensities by binning test intensities by their feature's training median and reporting the average error per bin. For protein groups on both development datasets the minimal median intensity was 23 for protein groups (**Fig. 2c-d**). The average error for intensities was roughly twice as large for intensities from the minimum median intensity bin in comparison to the overall performance (**Fig. 2e-f**). Interestingly, worst performance was observed for protein groups with the highest observed median intensity, which for intensities from one or six protein groups had a two to four times worse MAE than the overall MAE. The relative performance between the models was consistent across the bins. "

line 52: I suggest expanding the description to the two categories, missing not at random (MNAR), e.g. LOD related missingness, and missing completely at random (MCAR) where we do not know any systematic reason.

We added a sentence describing the two categories:

Lines 46-49: "In general, the community differentiates between missing at random (MAR) which is assumed to affect all intensities across the dynamic range, whereas missing not at random (MNAR) becomes more prevalent the more the intensity of a peptide approaches the limit of detection of the instrument."

line 63: This also depends on the instrument and the search engine used for peptide identification.

We changed the description:

Lines 58-62: "The noise in data from the instrument as well as peptide identification is most abundant for label-free quantification proteomics in DDA with missingness ranging from 10-40%¹², but for instance, blood plasma measured using DIA in a study of ALD still contained 37% missing values across all samples and protein groups before any filtering."

line 72: Suggest to use the term log-normal distributed.

We now use to the term log-normal:

Lines 71-74: “The protein intensities, stemming from aggregation of the precursor and/or fragment ion values in MS1 and MS2 scans, are assumed to be log-normally distributed, i.e. the log transformed intensities are entirely determined by their mean and variance.”

line 86: Why do the authors argue that it is not possible to re-use models from scRNA data? scRNA-seq data is typically modeled using a negative binomial distribution computed as a poisson-gamma mixture, which is a continuous distribution. Similar to other distributions, a linker function can be applied (here log transformation) to get something closely related to a gaussian distribution. This principle is used in existing autoencoders for MVI in scRNA-seq data. This could also work for single cell proteomics, as long as there is enough data available.

Thanks for the hint. We only encountered count-based autoencoders (Eraslan et al. 2019; Lopez et al. 2018), which assumes count distributions. These will in their limit also get closer to normal distributions, but depending on the number of reads this might or might not be true. We removed the statement from the introduction.

line 127: I suggest rephrasing this as the 49 proteins found to be significantly differentially expressed may only be due to imputation, that doesn't mean that these are significant due to uncovered biology through the imputation. How many proteins would have been associated with fibrosis when randomly selecting proteins from the list of not differentially expressed ones?

The ALD analysis was changed based on your above comment. The summary sentence in the abstract now reads as:

Lines 28-31: “When analyzing the full dataset we identified 27 additional proteins (+11.8%) that are significantly differentially abundant across disease stages compared to no imputation and found that some of these were predictive of ALD progression in machine learning models.”

The list of proteins was tested for association with fibrosis, one by one. The exact setup is described in the Methods section:

Lines 696-702: “The differential analysis was done using an analysis of covariance (ANCOVA) procedure using statsmodels (v.0.12) and pingouin (v.0.5)^{62,63}. We used a linear regression with the original kleiner score⁴⁷ as the stratification variable of interest for the patient's cirrhosis disease stage to predict protein quantifications, controlling for covariates. Therefore, effects for each protein group were based on an

ANCOVA controlling for age, BMI, gender, steatosis, and abstinence from alcohol as well as correcting for multiple testing as done in the original study.”

In general, the figure labels and text are often very small. I suggest increasing the size for most of them.

Thank you for the suggestion, we have now set font size 5 everywhere and export plot directly in the correct size, so there should be no more small fonts due to rescaling.

Formulas for loss function: Please explain all individual characters and symbols in all formulas, e.g. “z”.

The superscript z denotes parameters of the encoding space. We updated the paragraph with naming the sub-, and super-scripts.

In methods, the authors reference something with (ref. 24). Please remove the “ref.”. Please also use the same font and font size throughout the manuscript.

Thank you for the hints. We removed “ref” and unified the font and sizes.

I do not agree with the proposal that the data used for development will only be provided upon acceptance. Please upload and make the data available for the next revision.

We are sorry that the dataset was not available. We provided it a week after initial submission on the 15. February 2023 after we received the reporting schema by the editor. The data is released in a companion paper on PRIDE, see preprint. Additionally, we added a link in the Data Availability section to an extended HeLa dataset and the companion data descriptor paper.

The second to last paragraph in the Conclusion is very speculative and no evidence for most of the statements is provided in the manuscript. I suggest removing this.

We see the point that we speculate here based on our currently provided analysis. We removed the last two sentences of the second to last paragraph.

Please replace the references to bioRxiv by the respective peer-reviewed publications.

We apologise for the mistake and have updated the references.

Since best average compares very similar to the best model for each feature, I suggest only using the best average model to decrease complexity and reduce the redundant representation of 2e and 2f.

We removed the Figures 2e and 2f as suggested. We re-organized the workflow and a good model is now selected based on the validation data performance on the grid search.

Lines 675-680: “We picked the smallest model in terms of parameters of the top 3 performing ones as their performance was nearly equal on the validation data split. Besides the best models we reported results using other plots. The intensities in a split were binned by the median of the feature, e.g. protein group, they originated from based on the training data split. The MAE per bin was then reported (Fig. 2e,f). This is accompanied by a plot showing the proportion of missing values of a feature based on its median value over samples (Fig. 2c,d).”

Reviewer #3 (Remarks to the Author)

This submission focuses on a timely and important subject: missing value imputation in label-free quantitative proteomics (LFQ). To tackle it, the authors propose an original path, based on deep learning. To avoid a loss of generalization power (owing to the diversity of proteomics data and to the amount of data necessary to train over-parametrized models), the authors leverage a self-supervised strategy (where the predictive model is trained in a supervised way using automatically generated labelled and masked data, as classically done in computer vision). In addition, the proposed strategy has many interesting features, like the capability to process large cohorts or to impute at different levels (ions, peptides, proteins).

Whilst I am positive about the approach, the current version of the article failed to convince me on three aspects:

1. The practical constraints of the approaches are not presented, evaluated or discussed.
2. There seems to have flaws in the evaluation setting.
3. The comparisons against pre-existing methods are not sufficient.

These elements, detailed below, must be addressed before considering a publication.

Details:

1- The authors must be more explicit about the applicability of their approaches, notably for an interdisciplinary venue like Nature Communications. In the current state, the reader has no information

about the computational load and the execution time (including the self-training step) and how these scale depending on the data size (numbers of features, samples, MVs, etc.).

This is a fair point. We have now added information about the compute requirements and runtime. In general the models are fast and scale to large sample sizes, for instance, the entire DAE training and inference for precursors on the large dataset took 4min and 40 seconds, whereas random forest did not complete.

Runtimes are presented in Supplementary Data 4 and we further added a sentence in the main text:

Lines 194-200: “Runtime for imputation scripts for protein groups including data loading, manipulations and training were between 24 seconds and 52 mins (KNN: 0:24, VAE: 0:53, DAE: 1:30, CF: 2:21, RF: 51:54) using a T2000 GPU with 4GB of memory for the PIMMS models (**Supp. Data. 4**). Runtimes of PIMMS models for the precursor were higher (Median: 1:34, KNN: 2:30, DAE: 4:40, VAE: 5:16 and CF: 16:12). In general runtimes varied based on the number of epochs, mini batch-size, model architecture, patience for early stopping and the initial random weights”

In Supplementary Data 4 we listed the execution time for the all datasets for the three data levels. The total trained epochs depends when the early stopping criteria was met. Although the architectures are not 100% equivalent, the table gives a rough indication of how the number of features influence the execution time. The times here are produced using my personal machine using one core of an Intel i7-2.7 GHz and a built in Quatro T2000 GPU (4GB RAM), using the h:min:sec format

method	evidence	evidence N=50	peptides	peptides N=50	Protein Groups	Protein Groups N=50	ALD Protein Groups
CF	0:16:12	0:01:06	0:11:03	0:01:09	0:02:21	0:00:32	0:00:39
DAE	0:04:40	0:00:52	0:04:46	0:01:11	0:01:30	0:00:46	0:00:30
KNN	0:02:30	0:00:33	0:02:12	0:00:31	0:00:24	0:00:15	0:00:13
MEDIAN	0:01:34	0:00:23	0:01:25	0:00:22	0:00:21	0:00:16	0:00:15
COLMEDIAN	0:01:08	0:00:14	0:01:06	0:00:16	0:00:17	0:00:09	0:00:11
KNN_IMPUT E	0:01:45	0:00:20	0:01:53	0:00:18	0:00:21	0:00:17	0:00:10
MINDET	0:01:09	0:00:15	0:01:13	0:00:19	0:00:15	0:00:13	0:00:10
MINIMUM	0:01:02	0:00:15	0:01:14	0:00:14	0:00:15	0:00:09	0:00:08
MINPROB	0:01:19	0:00:15	0:01:15	0:00:15	0:00:16	0:00:21	0:00:13
PI	0:01:06	0:00:15	0:01:11	0:00:14	0:00:20	0:00:18	0:00:09
QRILC	0:02:16	0:01:02	0:02:12	0:00:57	0:00:21	0:00:29	0:00:13
ROWMEDIAN	0:01:21	0:00:20	0:01:09	0:00:16	0:00:16	0:00:10	0:00:09
ZERO	0:01:23	0:00:17	0:01:16	0:00:13	0:00:14	0:00:11	0:00:09
VAE	0:05:16	0:01:04	0:03:34	0:01:07	0:00:53	0:01:37	0:00:31
IMPSEQROB		0:00:42		0:00:36	0:00:23	0:00:15	0:00:11
LLS					0:05:33	0:08:03	0:00:23

MLE		0:01:28		0:01:42	0:00:20	0:00:16	0:00:09
RF					0:51:54	0:04:58	0:01:00
SVDMETHOD		0:01:11		0:00:55	0:47:31	0:00:13	0:01:47
BPCA		0:20:44		0:18:36		0:02:18	0:18:27
MICE-NORM		0:04:42		0:03:34		0:00:42	
MICE-CART						0:07:43	
IMPSEQ						0:00:26	
IRM						0:04:17	0:05:06
SEQKNN						0:00:20	

Using the small development dataset (protein groups), running the training notebooks on github workflow runner for windows takes (see specs: 2-core CPU (x86_64), 7 GB of RAM, 14 GB of SSD space):

Dataset\method	CF	DAE	VAE	Median	KNN (sklearn)
50 samples, 4,405 protein groups	19 sec	54 sec	50sec	7 sec	8 sec

Runtime issues will decrease as more hardware has specialized chips for vector operations (either GPUs or other like M1 chips).

Similarly, most of LFQ experiments relies on a handful of samples within each biological group (3 to 10), making the imputation both difficult and necessary. Conversely, the authors evaluate their methods on datasets with 50 to 450 samples, for which the imputation can be expected to be easier and not as necessary. Thus, it is currently impossible to assess whether it makes sense to apply one of the proposed methods to a "classical" LFQ dataset. As any method, this one must have limitations, and it is not inherently a problem. However, it is important to clearly formulate these limitations (and to support them experimentally), as they will dramatically influence the concrete interest of Nat Comm readership.

We very much appreciate the comment and have performed analyses of our and the additional methods for a wide variety of sample sizes. These results show that the deep learning based methods have the best performance on the 500 sample HeLa dataset, and are in top 5 for the 50 sample HeLa dataset only beaten by Bayesian PCA (BPCA). We performed analyses from 10-40 samples where we find CF to perform good for smaller sample sizes. Other methods show good result here as well, although especially on the higher dimensional data of peptides and precursors, all the deep learning methods are still in the top 5. Finally, for the clinical ALD dataset, the deep learning methods had best mean average error (0.42-0.44) on the test set where for instance BPCA had a MAE of 2.4. Importantly, the PIMMS framework allows one to test and apply the method that has the best performance on your own particular dataset for MAR values. These results can be found in Supplementary Data 3.

We updated the main text:

Lines 202-214: “Reducing the training sample by roughly a tenth, performance on the smaller development dataset (N=50) was comparable to the performance on the larger development dataset, while runtime reduced by 5 to 15 times for precursors (Median: 0:23, KNN: 0:33, VAE: 1:04, DAE: 0:52, CF: 1:06). For the smaller development dataset on the protein groups level, additional R methods from NAGuideR were able to complete. We found that BPCA³⁶ performed best on the 50 homogenous HeLa samples with relatively long runtime (e.g. on precursors: BPAC: 20:44, MICE-NORM: 4:42). For 10, 20, 30 and 40 samples, other methods showed good results, although PIMMS models still fit the data (**Supp. Data 3**): For example with 10 samples the precursors were fitted best with an MAE of 0.57 using BPCA and 0.61 using CF. In summary, this indicated that on an unnormalized, intensity varying dataset from a single machine the PIMMS models were able to capture patterns between detected features to impute values (**Supp. Fig. S3a**) for as few as 50 samples with competitive performance to other state-of-the-art methods.”

2- The simulation experiments (to estimate the MAE) are not sufficiently described. From my understanding, the masked MVs have been randomly chosen as a result of a classical train/validation/test split. However, doing so results in only generating MCAR values for subsequent evaluation. MCAR values can be efficiently imputed using pre-existing methods (e.g. random forest -- <https://www.nature.com/articles/s41598-021-81279-4>), but on the other hand, they are not realistic (LOD related MVs are not MCARs). To fairly assess the performance of an imputation method, it is necessary to compare the MAE on masked datasets with a varying ratio of MCAR/MAR (like in refs. 13 or 17).

The experimental design to evaluate different shares of MCAR/MAR seems to be based on an abundance of quantified features. A threshold matrix is defined (for mentioned example here) and this one is then used to simulate MNAR, assuming that lower abundant features are rather MNAR. A two-step experiment is performed for selection into training and test split. First, based on the threshold matrix which has higher probabilities (thresholds) for lower intensities, a Bernoulli draw decides if a feature is masked. Second, MAR are added by randomly selecting intensities from the entire matrix. As we need to learn from intensities, removing more intensities from rarely present features sounds like a penalty for machine learning focused models. We could however envision to augment less frequent values in our training procedure.

In order to allow an easier assessment of MCAR/MAR, we changed the comparison plot across the range of intensities. We binned the intensities in the test split by the median's integer floor value (i.e. rounded down to nearest integer) of the training data and then the models are compared per bin using the MAE of the intensities in that bin. This shows in Fig. 2e,f and Fig. 4c giving the number of intensities in a bin in parentheses in the x-axis labels. Fig. 2 c,d show the proportion of missing values per protein group binned again by the median's integer values. Comparing 2c,d to 2e,f indicates the performance for the interesting ranges of intensities per method. Overall for a new dataset the MCAR to MNAR fractions are

unknown. We don't have an answer to how to evaluate a dataset the best way possible, however we rather evaluate performance using MCAR simulated missing values using the proposed plots.

Likewise, the ALD experiment, though interesting, is hardly convincing: the proposed approach allows to discover more differentially abundant proteins, including a handful of biologically relevant ones, but the discussion does not account for the false positives. On the other hand, when this information is accounted for (like in fig 3e) the interest of the proposed approach seems marginal, notably at low FPR.

To investigate this we performed differential analysis without imputing any features to the ALD experiment. It's only a proxy for biological relevance, but our results indicate that RSN imputation can lead to hiding biological signals which should not happen using imputation: Some protein groups are not differentially abundant when RSN imputations are added and others are only differentially abundant with these (Supp. Data. 10). We observed out-of-distribution imputations for these protein groups (Fig. 4 d-e). Additionally we added an experiment where we removed 20 % of the data at random, leading to 22 protein groups not being differentially abundant without any imputation. We then compared how many lost differentially abundant protein groups could be recovered using the models, showing that CF, DAE and VAE could recover most of the lost signal due to added MAR missing values (VAE: 20, DAE: 20, RF: 20, CF: 16 of 22):

Lines 308-325: "Then we tested the ability of the methods to recover differentially abundant protein groups when removing data at random (MAR). Therefore, we randomly removed 20 percent of the data. Comparing the results of the differential abundance analysis using the complete dataset compared to the reduced dataset we observed a decrease in the number of differentially abundant protein groups from 212 to 192 (ANCOVA, Benjamini-Hochberg multiple testing correction, $p\text{-value} \leq 0.05$). Interestingly, we found that besides 22 protein groups losing their differential abundance, four became differentially abundant by randomly removing data. To assess the effectiveness of various imputation methods, we used no imputation as a benchmark. We found that DAE, VAE, and RF imputation techniques successfully recovered 20 out of the 22 protein groups that lost their differential abundance. In contrast, median imputation only recovered two protein groups, while CF imputation recovered 16 groups with three protein groups slightly above the cutoff of 0.05. KNN recovered 13 protein groups (**Fig. 3c,e, Supp. Data 8**). Four protein groups became differentially abundant without imputation when intensities were removed in the 80% dataset, which we labeled as false positives to follow our approach (**Fig. 3d,f**). However, three of the four were nearly statistically significantly abundant in the 100% dataset and therefore adding back patients through imputation to the analysis can be seen as beneficial. This indicated that the PIMMS models were capable of recovering lost signals of at least MAR data."

Former Fig 3e, now part of Supp. Fig. S8, showed that the new proteins also have predictive power, but indeed, for this study a classifier could already be built with the initial set of stably quantified protein groups.

3- The anterior literature is not adequately accounted for. Although all the relevant articles are referenced, they are not correctly cited/contextualized. Notably, according to the authors, proteomics imputation is essentially conducted at protein level with the assumption that missing values result from a detection limit (like with the default Perseus pipeline), but the conclusions of refs 12-18 are more nuanced (to put it mildly) and these should have been considered in the experimental evaluations. Concretely, the authors compare their 3 methods against 3 baseline methods: median imputation (a notoriously poorly performing method), interpolation (which is not an imputation method per se, as a time series structure must be assumed, contrarily to most of the other methods) and RSN (not applied to the HeLa dataset). However, a host of alternative (and more refined) approaches have been published, each of them performing differently depending on the amount of MVs, their MCAR/MAR nature, the imputation level (ion, peptide or protein), etc. For instance, the NAGuideR paper (ref 18) lists 15 to 20 methods and proposes a unified framework to compare them. It is of the utmost necessity the authors comprehensively evaluate on both datasets (HeLa and ALD) their methods against all the state-of-the-art methods, not only a couple of notoriously poor performing ones.

Thank you very much for the suggestion. We used the NAGuideR paper and tool as a guide for selecting methods to benchmark against. As NAGuideR is a ShinyApp that bundles logic and graphical user interfaces we adapted it to run as a script in the comparison workflow. However, for various reasons some packages failed to run without any information about the errors. We were successful in integrating all methods listed in Table S1 in Wang et al. 2020, except GRR and GMS which we did not manage to install in our workflow and compared the methods in a reproducible environment (**Supp. Table 1**). However, for some datasets not all methods completed due to undetermined execution after more than half an hour (also running for several hours did not yield results), or reasons which are not made explicit by the raised error messages from the R packages. An overview of the available methods in the snakemake workflow was added to the README of the package and in **Supp. Table 1**. The available methods that ran on all our datasets are (using the NAGuideR names): ZERO, MIN, COLMEDIAN, ROWMEDIAN, KNN_IMPUTE, QRILC, MINDET, MINPROB, PI. On medium to large scale protein group datasets (up to several hundred samples, and thousands of protein groups): RF. See an overview of performance and runtimes in **Supp. Data 3, 4**. As mentioned above, the results of these analyses show that the introduced models have a competitive performance, especially CF for smaller sample sizes.

To summarize, the authors proposed a new appealing approach to LFQ imputation, but in the current state of the manuscript, the presented experimental evaluations are not sufficient to demonstrate its practical applicability (point 1), its accuracy in various realistic MV settings (point 2) and how it performs with respect to pre-existing methods (point 3).

By adding additional methods to the comparison next to runtimes, a plot of performance per median of features the simulated missing values originated from and a recovery analysis after introducing 20 percent MAR values, we hope that we have been able to address the comments of the reviewer.

References

- Allesøe, Rosa Lundbye, Agnete Troen Lundgaard, Ricardo Hernández Medina, Alejandro Aguayo-Orozco, Joachim Johansen, Jakob Nybo Nissen, Caroline Brorsson, et al. 2023. "Discovery of Drug-Omics Associations in Type 2 Diabetes with Generative Deep-Learning Models." *Nature Biotechnology*, January. <https://doi.org/10.1038/s41587-022-01520-x>.
- Eraslan, Gökcen, Lukas M. Simon, Maria Mircea, Nikola S. Mueller, and Fabian J. Theis. 2019. "Single-Cell RNA-Seq Denoising Using a Deep Count Autoencoder." *Nature Communications* 10 (1): 1–14.
- Lopez, Romain, Jeffrey Regier, Michael B. Cole, Michael I. Jordan, and Nir Yosef. 2018. "Deep Generative Modeling for Single-Cell Transcriptomics." *Nature Methods* 15 (12): 1053–58.
- Nissen, Jakob Nybo, Joachim Johansen, Rosa Lundbye Allesøe, Casper Kaae Sønderby, Jose Juan Almagro Armenteros, Christopher Heje Grønbech, Lars Juhl Jensen, et al. 2021. "Improved Metagenome Binning and Assembly Using Deep Variational Autoencoders." *Nature Biotechnology* 39 (5): 555–60.
- Wang, Shisheng, Wenxue Li, Liqiang Hu, Jingqiu Cheng, Hao Yang, and Yansheng Liu. 2020. "NAGuideR: Performing and Prioritizing Missing Value Imputations for Consistent Bottom-up Proteomic Analyses." *Nucleic Acids Research* 48 (14): e83.

Reviewers' Comments:

Reviewer #2:

Remarks to the Author:

I appreciate the efforts of the authors to address all of the major issues we pointed out, that additional experiments and comparisons were conducted, access to the data was provided, and supplemental data was added.

I also appreciate the fact that it was made more clear by the authors that the problem addressed is missing value imputation within one dataset, with models specifically trained for individual datasets to capture the underlying distribution that is specific to a certain biology, tissue type or mass spec device without the goal of generalizing to unseen data, this clarifies the scope and objectives.

The fact that one cannot (currently) differentiate between "true zeros" is potentially preventing application in more general clinical setting. While this is a limitation of the work, I acknowledge this is a general limitation and accept that there is no clear way of addressing these issues at this point in time, primarily due to technological limitations of mass spectrometry itself.

In my view, the major contribution and novelty in this work is the investigation of missing value imputation in the MAR case on different levels of aggregation. However, a large number of minor issues remain which need to be addressed before recommending publication:

- I suggest revising most figures with respect to font sizes used. Also, I believe the figure legends would benefit from a second round of revision to enable a more general readership to fully understand what is shown.

- Figure 1:

o Panels b/c, it is not clear what the color indicates, a legend is required.

- Figure 2:

o c/d description is very difficult to understand. As far as I understand it, it is showing two things.

1. It shows the distribution of protein group prevalence values (y-axis), where prevalence is the relative number of non-zero intensity protein groups within one sample. The overall distribution is then stratified by grouping samples into bins of size 1 using their median protein intensity (x-axis). It can be seen that there are many samples with a low median protein intensity and a large spread of protein prevalence with generally higher missingness and lesser samples with a lesser missingness with increasing median protein intensity. This also makes sense, since the expectation is that overall higher protein intensity means that many low abundant proteins are still above the LOD. But it is also showing that missingness is not solely to MAR but rather MNAR. Make the figure more clear by using a descriptive y-label, e.g. "percentage of zero intensity proteins in sample" and use "protein groups" instead of "features" it only shows protein groups.

o Something went wrong in the descriptions of the individual colors during the revision process, e.g. blue is now median but KNN at the same time and yellow does not exist anymore, also orange is kNN-impute or MICE-Norm.

o Please describe what N and M refers to.

- Figure 3:

o Legend appears very inconsistent, "80% dataset" (line 330) vs "80 percent" (line 331) and "80 dataset" (line 332-333).

o c/d Please add what the q-values refers to? This is missing from the figure description to be comprehensible.

o c/d Please add what the grey line representing?

o line 333: typo "Taking 22 the" should be "Taking the 22"

o Figure description for e and f is not descriptive, and appears more like an instruction.

o e/f Y-label is missing.

o e Please add what the what the model "v" in the third bar group is.

o I don't understand what "None" is and how it can have TN/FP? I also did not find description for this in the in the text.

o f: line 335: "labeling differentially abundant [xyz] as ..." xyz is missing, what should it be? Protein groups?

o I recommend using consistent coloring scheme for all models throughout the Figures. Yellowish-green color is called median here, but in Figure 2 it was BPCA.

o line 337: typo, it is "not differentially abundant", you wrote "non".

- Figure 4:

o b I recommend using a consistent description with that of Figure 2c/d.

- o d-g: Dots and crosses are indistinguishable from each other.
- o What are the q-values, needs to be in the legend.
- I also recommend carefully double checking the supplemental figures for similar issues wrt consistency, font sizes and self-explanatory legends.
- line 267, line 274: precursors, peptides and protein groups do not “perform”, that does not make sense. “models perform ... for e.g. peptide level”
- line 408: typo, “median” shouldn’t be capitalized
- line 444-446: What is a “protein groups median imputation”? If I interpret correctly, maybe write “Using median imputation for all available protein groups yielded ...”
- Please carefully review the section from line 404 – 425, these are a few things I noticed:
 - o line 414: The models don’t change the result n times, “The results don’t change when running the models 10 times”
 - o line 414-416, style, maybe rewrite to “A total of 46 out of 64 newly included protein groups ... were ...”
 - o line 417: grammar and unclear
 - o line 418, comma is wrong, i.e. “Additionally, 13 ...”
- As a last remark and a suggestion to further clarify this in the discussion, the authors response to the question about generalization from the first review, they clarified things and pointed to line 506-515 where it is brought up how to use a fine tuning approach rather than trying to come up with a model for generalization. The authors state that “feature embeddings could be transferred and extended easily” (line 511) and “similar cohorts, e.g. for the same tissue and similar patients” (line 513) could be “the basis to build a database of tissue specific models”. While I generally agree, this does not address batch effects introduced by artifacts e.g. differences in sample preparation, acquisition, MS, etc. This technical aspect is not discussed and may actually limit further progress on this topic, limiting the integration of large amounts of heterogeneous mass spectrometry data from various labs or machines into a broad dataset for training better models.

Reviewer #3:

Remarks to the Author:

While I was optimistic in my first review, I have to advise the editor to reject this submission, essentially because it does not comply with scientific publication good practices. Concretely, the authors:

- 1- Keep ignoring a significant portion of the state of the art (see details below).
- 2- Do not sufficiently delineate their application scope and overstate it (see details below).
- 3- Present biased comparisons and statistically incorrect results/interpretations (see details below).

Correcting for these points would be possible, but a fairer presentation would lead to a low-impact publication. Indeed, after digging into the supplemental materials and the uncited literature on the subject, it looks clear to me the authors have developed three imputation methods that only make sense on a very specific problem (datasets with a hundred+ of samples where missing values occur completely at random). Moreover, on this restricted setting (which is the one where imputation is the less challenging and for which many solutions already exist), the performances exhibited are at best, correct only (ie. better than naive methods, yet not as good as the best ones currently available).

Please find below more detailed explanations:

1- State of the art incompleteness:

In my first review, I mentioned NAGuideR (both a review and an evaluation platform) to pinpoint the enormous lack of reference methods (the original version only referred to median imputation, to RSN a Perseus-like method, and to an ad-hoc interpolation method). In the revised version, the authors cited NAGuideR but did not look any further and missed recent but major innovations in missing value imputation, notably msImpute [PMID: 37105364], the last version of MsStats [<https://pubs.acs.org/doi/10.1021/acs.jproteome.2c00834>], ProJect [PMID: 37419612], less recent ones like GSimp [PMID: 29385130], as well as possibly others (given the dynamics of the field). It is important the authors read the bibliography before claiming for an innovation.

2- Application scope:

In my first review, I insisted on the need to test the PIMMS methods in realistic scenarios, with a varying proportion of MNARs. The authors declined my recommendation using dubious arguments (see next section). On the other hand, the material they present gives evidence their methods are MCAR only. Notably, the observed trend of imputing LOD MVs nearer to the distribution mode in a "conservative" way (see lines 476-478) is a feature common to most of MCAR methods. Similarly, the construction of the train/test split is in general associated to MCAR imputation (which is also the reason why RF works well essentially on MCARs). In itself, proposing an MCAR-only algorithm is not a problem, however:

- MCAR values are the easiest ones to impute and many methods already do so.
- The article never explicitly points this restriction to MCARs, while on the contrary, it makes a big deal of discussing the MAR/MNAR dichotomy, in the introduction, in the results and in the discussion. Sometimes, this argument is also used (see line 192) to discard imputation methods from comparisons, like RSN (there are datasets on which simple MNAR methods like RSN or other work well, notably for biomarker discovery). Worse, in sentences like those of lines 476-478, the authors present a general feature of MCAR methods as if it was specific to their approach, which is misleading.
- Line 307 and after, the authors claim they can impute MAR (ie any MV that is not MNAR, not only MCAR ones), while they do not demonstrate it. Unless a specific (not shown) algorithm is used, randomly removing 20% of the data does not simulate MARs in general, but MCARs. Imputing MAR values with MCAR algorithms is a common shortcut in proteomics as there is no better way, however, it should not lead the authors to overstate their MAR imputation capabilities. Put together, these elements give the impression the authors overstate their application scope.

3- Biased comparisons and statistically invalid results:

This is probably the most important issue. Although the main text gives the impression the proposed methods perform reasonably well (and often a bit better than their competitors), a deeper look into the supplemental materials tells another story:

- If one looks at NAGuideR's article as well as recent other articles, the 3 best methods seem to be ImpSeq(rob), SeqKNN and BPCA. Conversely, KNN and RF, although efficient on theoretical datasets with MCARs only (see my first review) do not appear to be accurate on real datasets where MARs and MNARs coexist. Knowing this, the prevalence of KNN and RF in the main article is rather puzzling (BPCA and SeqKNN only occasionally appear, as well as MICE-NORM, RSN or MEDIAN).
- A similar bias appears in the selection of MNAR-only methods: TrKNN never appears and does not seem to have even been tested according to Supp Data 4 (while it provides rather good performances according to NAGuideR) and among the many other ones tested (MINIMUM, MINDET, MINPROB, QRILC, ZERO and PI, the latter being equivalent to RSN), QRILC systematically outperforms RSN/PI according to Supp Mat 3, so using RSN as benchmark looks like a convenient bias.
- In Supp Data 3 (lower table), it appears that with experiments of small to intermediate sizes ($n < 50$), ImpSeq, SeqKNN and BPCA outperform the proposed methods and the other NAGuideR methods. In this context, lines 208-214 are so misleading they can be considered as statistical malpractice. Notably, sentence like "For example with 10 samples the precursors were fitted best with an MAE of 0.57 using BPCA and 0.61 using CF" conveniently hide that in the first column of the table (ProteinGroups_N10), the best PIMMS method displays an MAE 25% larger than ImpSeq. Similarly, in the 4th and 5th columns, the best PIMMS method is not in the Top 3. Considering these numbers, calling any of the PIMMS method competitive requires an important mind twist.
- Likewise, on the upper table of Supp Data 3, ImpSeq has almost never been tested, while it is computationally rather efficient (at least far more than BPCA), which raises questions.
- Even if on the largest Hela dataset, state of the art methods may not scale up, the BPCA performances on the smaller dataset are such that a good imputation strategy would be to apply it batch-wise (in other words, it reaches similar performances than the PIMMS methods with fewer samples). I understand the scalability argument is of interest, but throwing away any method that cannot be fast on a set of 500 samples is a bit excessive. Let us recall that in their original submission, the authors used interpolation as a reference method, which is a KNN-like algorithm applied to a batch of size 3. Likewise, why not comparing state-of-the-art methods (notably BPCA or ImpSeq) on 10 batches of 50 samples? (assuming ImpSeq can indeed not scale, a fact I remain very skeptical about).

- Line 640, The KNN algorithm has not been used with the default tuning in most proteomic applications ($K=10$), but with $K=3$, which is likely to artificially lower its performances. Conversely, it appears the HeLa dataset has been used for hyperparameter tuning (Supp Fig 1) as well as for evaluation. Owing to the medium-only stability of DAE and VAE (Supp Fig 1), I fear data overfitting (already mentioned by another reviewer) has artificially boosted the performances displayed. As a result, I suspect the performances displayed are optimistic for the PIMMS methods and pessimistic for the reference methods (notably KNN).

- In most of the panels from figures 2 and 3, the authors compare the 3 PIMMS methods to 3 (or less) state of the art imputation methods (generally KNN and RF). In those figures, the best PIMMS method is not always the same (eg it is DAE on 2A and CF on 2B), which can make sense. However, claiming the efficiency of PIMMS approach in general (ie including the 3 different algorithms) as done line 214 or in the discussion (lines 466, 476, 484, etc.) is not correct, as doing so would require adding a multiple test correction. Otherwise, one just has to propose a toolbox with 97 random methods, to compare them with 3 reference ones, and it will be easy to claim that in 97% of the case, one of the proposed method outperforms the reference methods. Therefore, in the current context, the performances are exhibited in a statistically invalid way.

- Dubious arguments can also be found in the response to reviewers. Notably, claiming a varying ratio of MCARs/MNARs would penalize ML-based models is nonsensical (recall that KNN or RF are also based on training, and such imputation tools have long been evaluated that way). Similarly, assuming binning the MAE according to the intensity (Fig. 2E, 2F, 4C) is of interest to assess the MCAR/MNAR imputation capability does not hold for many reasons, the most blatant one being that in those figures, no MNAR method (like RSN) is compared to!

Overall, the amount of omissions, distortions, statistical approximations and biases is so important that the main text seems to describe an "alternative truth" with respect to the numbers and statistics available in the supp mat or in the literature.

Conclusions and recommendations:

Although the originality of the approach is worthwhile, the article cannot be published in its current form, because it steps away from scientific publication good practices (acknowledgement of prior literature, no overstatement/unsupported claim and most importantly, fair comparisons and statistically valid performance assessment). If the authors correct the manuscript accordingly, I think it will be publishable, but I fear it will no longer meet the expectations of a journal like Nat Comm. More precisely, I suspect a fair presentation of the results will show that the 3 presented algorithms are able to impute MCAR-only MVs on very large-only datasets with acceptable-only performances, ie not as good as ImpSeq, SeqKNN, BPCA or msImpute. Nevertheless, their easiness to scale-up may be of interest for some specific applications, so that a publication in a 2nd tier journal would make sense.

Point-by-point response

We thank the reviewers for their time and input to our manuscript. We have tried to meet all the requests from the reviewers, the most significant one has been to redo all simulation analyses with MNAR instead of only MCAR. Additionally, we want to highlight that we now implemented a total of 30 imputation methods into our framework which all can be accessed by a user. This means that the framework is not restricted to the self-supervised learning methods that we present, but rather a comprehensive set that can all be applied to a user's data. Please see below for our point-by-point response to the reviewers.

Reviewer #2 (Remarks to the Author):

I appreciate the efforts of the authors to address all of the major issues we pointed out, that additional experiments and comparisons were conducted, access to the data was provided, and supplemental data was added.

I also appreciate the fact that it was made more clear by the authors that the problem addressed is missing value imputation within one dataset, with models specifically trained for individual datasets to capture the underlying distribution that is specific to a certain biology, tissue type or mass spec device without the goal of generalizing to unseen data, this clarifies the scope and objectives.

The fact that one cannot (currently) differentiate between “true zeros” is potentially preventing application in more general clinical setting. While this is a limitation of the work, I acknowledge this is a general limitation and accept that there is no clear way of addressing these issues at this point in time, primarily due to technological limitations of mass spectrometry itself.

In my view, the major contribution and novelty in this work is the investigation of missing value imputation in the MAR case on different levels of aggregation. However, a large number of minor issues remain which need to be addressed before recommending publication:

Thank you a lot for the detailed corrections. Having feedback on what is unclear in e.g. figures is very valuable and a key to improvements!

- I suggest revising most figures with respect to font sizes used. Also, I believe the figure legends would benefit from a second round of revision to enable a more general readership to fully understand what is shown.

We increased the font size from a minimum of 5 to a default of 6. Some figures now have an even larger font size (up to 8) in the legends. All main figures and most supp. Figures are hopefully now better to read.

- Figure 1:

o Panels b/c, it is not clear what the color indicates, a legend is required.

The heatmap represents the number of identified features in a sample. We added a legend.

- Figure 2:

o description is very difficult to understand. As far as I understand it, it is showing two things. 1. It shows the distribution of protein group prevalence values (y-axis), where prevalence is the relative number of non-zero intensity protein groups within one sample. The overall distribution is then stratified by grouping samples into bins of size 1 using their median protein intensity (x-axis). It can be seen that there are many samples with a low median protein intensity and a large spread of protein prevalence with generally higher missingness and lesser samples with a lesser missingness with increasing median protein intensity. This also makes sense, since the expectation is that overall higher protein intensity means that many low abundant proteins are still above the LOD. But it is also showing that missingness is not solely to MAR but rather MNAR. Make the figure more clear by using a descriptive y-label, e.g. "percentage of zero intensity proteins in sample" and use "protein groups" instead of "features" it only shows protein groups.

The y-axis gives the prop. of missing values of a protein group. We adjusted the x-axis and legend.

We updated the legend to the following text:

Lines 295-297: "**c,d**) Protein groups medians were binned by their integer median value and the boxplot of the proportion of missing values per protein group is shown for large and small development. Number of protein groups in bin in parentheses."

o Something went wrong in the descriptions of the individual colors during the revision process, e.g. blue is now median but KNN at the same time and yellow does not exist anymore, also orange is kNN-impute or MICE-Norm.

We did not keep the colors consistent between revisions but ensured that for the core methods they were the same across all figures and analysis. We now extended the range of possible colors, which hopefully avoids color duplications.

o Please describe what N and M refers to.

We apologize. N was the number of samples and M was the number of protein groups, peptides or precursors. We now add it to the legend of Figure 2a.

Lines 285-294: "**a**) Performance of best imputation methods at the level of protein groups, aggregated peptides, and precursors for MaxQuant outputs with 25 percent MNAR. Mean averaged error (MAE) is shown on the y-axis. Blue: KNN (scikit-learn), green: CF, red: DAE, purple: VAE, olive: BPCA, brown: random forest (missForest), orange: KNN_IMPUTE(impute), pink: IMPSEQ, dark-orange: MICE-NORM, sand: SEQKNN. The DL methods performed better or equally good in comparison to other models. Performance was similar for all three models on each data level and less aggregated data, i.e. models on precursors and aggregated peptides performed better compared to models on protein groups. BPCA and RF did not finish for aggregated peptides and precursors within 24 hours. Models were ordered by

overall performance on the three datasets combining the best five for each. N: number of samples, M: number of features.”

- Figure 3:

o Legend appears very inconsistent, “80% dataset” (line 330) vs “80 percent” (line 331) and “80 dataset” (line 332-333).

We changed it to say 80% dataset in all these cases now.

o c/d Please add what the q-values refers to? This is missing from the figure description to be comprehensible.

We refer to these now as multiple testing adjusted p-values.

o c/d Please add what the grey line representing?

We added this now.

o line 333: typo “Taking 22 the” should be “Taking the 22”

Changed. Thanks.

o Figure description for e and f is not descriptive, and appears more like an instruction.

We changed the sentence order making it more descriptive.

o e/f Y-label is missing.

We added the label “count”.

o e Please add what the what the model “v” in the third bar group is.

That was the random forest (RF). We have updated this throughout the text and workflow.

o I don’t understand what “None” is and how it can have TN/FP? I also did not find description for this in the in the text.

We changed the x-axis label from “None” to “no imputation”. The bar indicated the reference group, i.e either all false negatives or all false positive comparing the results using the entire dataset without imputation to the reduced dataset without imputations.

o f: line 335: “labeling differentially abundant [xyz] as ...” xyz is missing, what should it be? Protein groups?

We added the missing noun, which was indeed protein groups.

o I recommend using consistent coloring scheme for all models throughout the Figures. Yellowish-green color is called median here, but in Figure 2 it was BPCA.

We now assigned unique colors that are consistent across the main figures.

o line 337: typo, it is “not differentially abundant”, you wrote “non”.

Changed.

Dealing with each suggestion above, the legend of Figure 3 now reads as follows:

Lines 346-360: “**Fig. 3: Performance on simulated missing data with 25 percent MNAR in ALD dataset and effects of imputation on differential analysis.** **a)** Performance for protein groups of plasma proteome data using a share of 25 percent MNAR simulated missing values on the full dataset and **b)** on the 80% dataset using the same configurations (LD: latent dimension, HL: hidden layer dimension). The performance of best five imputation methods for protein groups of plasma proteome data using a share of 25 percent MNAR simulated missing values (red: DAE, dark-green: truncated KNN (TRKNN), green: CF, brown: RF, purple: VAE) **c)** q-values, i.e. multiple testing adjusted p-values, for 17 protein groups which were differentially abundant using the full dataset without imputation, but not for the 80% dataset without imputation. The gray line indicates the five percent FDR cutoff. No imputation (None) shows the result without imputing values. **d)** q-values for three protein groups which were differentially abundant using the 80% dataset without imputation, but not initially when using all data. **e)** Count of false negative (FN) and true positives (TP) per method on the reduced dataset taking the 17 differentially abundant protein groups from c) on the full dataset as ground truth. **f)** Same as in e) for three protein groups in d), labeling differentially abundant protein groups in the 80% dataset as false positives (FP) since they did not show up as differentially abundant in the full dataset. The three examples are around the FDR cutoff without imputation. True negatives (TN) are not differentially abundant here (**Supp. Data. 5**). No imputation (None) is the reference defining the labels.”

- Figure 4:

o b I recommend using a consistent description with that of Figure 2c/d.

o d-g: Dots and crosses are indistinguishable from each other.

o What are the q-values, needs to be in the legend.

We updated the legend, increased the dot-size and font-size as well as added the q-value description to the legend.

The Figure 4 legend now reads as:

Lines 481-490: “**Figure 4: Underlying data distribution and performance in detail on an ALD dataset.** **a)** Stacked histogram of validation and test split using a share of 25 percent MNAR. Bins were created using the integer value of the log₂ transformed intensities. N is the number of intensities in the split, each corresponds to 5 percent of all available quantifications. **b)** Protein groups medians were binned by their integer median value and the boxplot of the proportion of missing values is shown. The number of protein groups in a bin is in parentheses. **c)** MAE with 95% confidence interval for protein groups intensities in the test split binned by the integer value of the protein group’s median intensity in the training data split. RF was replaced with QRILC to showcase a MNAR focused method. N: number of intensities in a bin from test split. **d-g)** Comparison of q-values, i.e. multiple testing adjusted p-values, based on no imputation (measured) and adding imputations using the models for four protein groups.

orange: kleiner score > 1, blue: kleiner score < 2, crosses in the first column indicate observed values, and dots in other imputation by models.”

- I also recommend carefully double checking the supplemental figures for similar issues wrt consistency, font sizes and self-explanatory legends.

- line 267, line 274: precursors, peptides and protein groups do not “perform”, that does not make sense. “models perform ... for e.g. peptide level”

This refers to the legend of Figure 2. For the fully corrected legend, please see above.

- line 408: typo, “median” shouldn’t be capitalized

Corrected.

- line 444-446: What is a “protein groups median imputation”? If I interpret correctly, maybe write “Using median imputation for all available protein groups yielded ...”

We changed the sentence according to the recommendation:

Lines 472-475: Using median imputation for all available protein groups yielded the best area-under the receiver operating curve (AUROC) of 0.90 compared to 0.86-0.90 for DAE, TRKNN, CF, RF, QRILC and VAE on the test samples (**Supp. Fig. S8**).

- Please carefully review the section from line 404 – 425, these are a few things I noticed:

o line 414: The models don’t change the result n times, “The results don’t change when running the models 10 times”

o line 414-416, style, maybe rewrite to “A total of 46 out of 64 newly included protein groups ... were ...”

o line 417: grammar and unclear

o line 418, comma is wrong, i.e. “Additionally, 13 ...”

We changed the paragraph titled “robustness of differently abundant protein group identification” according to the recommendations:

Lines 431-452: “Next, we decided to analyze in more detail which protein groups were differentially abundant using the different models and how reliable the differential outcome was per model. We repeated the analysis on the ALD data ten times with 25 percent MNAR simulated missing values, re-training the models on the same training data split. In this scenario median, TRKNN and RSN imputations were guaranteed to yield the same result for all repetitions. Of the 313 originally included protein groups, 27 did not have the same differently abundant outcome for all models on all ten repetitions. Especially for protein groups with few missing values, seeing the same outcome is not unexpected. However, for nine protein groups with two to 29 percent missing values, adding RSN imputations changed the result from being differentially abundant without imputation to being not differentially abundant. The results did not change running DAE, TRKNN, CF, RF and VAE ten times for these nine protein groups. A total of 46 out of the 64 newly included protein groups with a higher percentage of missing values were at least once differentially abundant by using one of the methods. Of

these 46 newly included ones, 16 were differentially abundant without imputation, which except by median or QRILC imputation was not changed using the top five model imputations. Additionally, 13 were at least identified seven times as differentially abundant using DAE, TRKNN, CF, RF or VAE imputations (**Supp. Table 2**). Using CF imputation, we found 17 protein groups to be differentially abundant in all ten repetitions. Deterministic TRKNN imputation in our setup found 28 protein groups to be differential abundant ten times. However, ten of these were only found using TRKNN imputations at least seven times (**Supp. Table 2**). We conclude that there is some variability in the analysis due to imputations and that for certain imputed protein groups the choice of the imputation method is crucial.”

- As a last remark and a suggestion to further clarify this in the discussion, the authors response to the question about generalization from the first review, they clarified things and pointed to line 506-515 where it is brought up how to use a fine tuning approach rather than trying to come up with a model for generalization. The authors state that “feature embeddings could be transferred and extended easily” (line 511) and “similar cohorts, e.g. for the same tissue and similar patients” (line 513) could be “the basis to build a database of tissue specific models”. While I generally agree, this does not address batch effects introduced by artifacts e.g. differences in sample preparation, acquisition, MS, etc. This technical aspect is not discussed and may actually limit further progress on this topic, limiting the integration of large amounts of heterogeneous mass spectrometry data from various labs or machines into a broad dataset for training better models.

Replacing batch effects from sample preparation, instrument settings and the type of acquisition is an open question. We would need to test the limits of learning from the data with or without encoding additional categorical information into a model. We normally impute the data after preprocessing, which if one would transfer models needs to yield comparable datasets - also if the analysis is performed without any imputation.

Lines 539-551: “In general, the modeling approaches here are restricted to the samples in a particular study and all models are fitted for each new dataset. However, transfer of models between datasets can be envisioned although a recent study suggested that this brings no benefits⁴². The potential to fine-tune a model trained on one dataset to a new dataset for a fixed set of features is possible without further efforts for autoencoders. For CF it would need to find the closest training samples in the case where samples are separated strictly into train, validation and test set. However, feature embeddings could be transferred and extended easily. Therefore, all models could potentially be envisioned in a clinical setup, where models are re-trained with the latest samples. This could be implemented using similar cohorts, e.g. for the same tissue and similar patients, which is then the basis to build a database of tissue specific models - or by incorporating tissue embeddings as an additional source of information. The difficulty in achieving this will be a stable setup for comparable results without major batch effects due to sample handling or different instruments. How to approach and the potential for data integration from different setups is an unresolved issue.”

Reviewer #3 (Remarks to the Author):

While I was optimistic in my first review, I have to advise the editor to reject this submission, essentially because it does not comply with scientific publication good practices. Concretely, the authors:

- 1- Keep ignoring a significant portion of the state of the art (see details below).
- 2- Do not sufficiently delineate their application scope and overstate it (see details below).
- 3- Present biased comparisons and statistically incorrect results/interpretations (see details below).

Correcting for these points would be possible, but a fairer presentation would lead to a low-impact publication. Indeed, after digging into the supplemental materials and the uncited literature on the subject, it looks clear to me the authors have developed three imputation methods that only make sense on a very specific problem (datasets with a hundred+ of samples where missing values occur completely at random). Moreover, on this restricted setting (which is the one where imputation is the less challenging and for which many solutions already exist), the performances exhibited are at best, correct only (ie. better than naive methods, yet not as good as the best ones currently available).

We, of course, do not agree with this assessment of our work. It was never our intention, as the reviewer suggests, to hide or in any way misguide the reader. I have in my career never received an assessment indicating scientific misconduct and want to emphasize that I value scientific integrity of the highest importance in my work (Simon Rasmussen).

Please find below more detailed explanations:

1- State of the art incompleteness:

In my first review, I mentioned NAGuideR (both a review and an evaluation platform) to pinpoint the enormous lack of reference methods (the original version only referred to median imputation, to RSN a Perseus-like method, and to an ad-hoc interpolation method). In the revised version, the authors cited NAGuideR but did not look any further and missed recent but major innovations in missing value imputation, notably msImpute [PMID: 37105364], the last version of MsStats [<https://pubs.acs.org/doi/10.1021/acs.jproteome.2c00834>], Project [PMID: 37419612], less recent ones like GSimp [PMID: 29385130], as well as possibly others (given the dynamics of the field). It is important the authors read the bibliography before claiming for an innovation.

We added three additional methods from two packages. The work was initially to present three new models to impute data - inspired by our large clinical cohorts. The request to compare to further possible methods now led to having both new methods and a comprehensive set of alternatives to pick from. We not only cited NAGuideR, but we also focused on making the packages referenced in NAGuideR available in our workflow. The workflow was tested continuously to investigate if the methods could still be installed and executed. Therefore, we require at least to have some sort of functionality published that can be either installed (preferred) or copied to our repository. In comparison to the review of R methods and evaluation platform NAGuideR, our workflow has a dedicated test set and does not use further annotations. Trying to run our datasets in NAGuideR failed for most of the methods and therefore we implemented these in our workflow to be consistent. Now there are a total of 30 methods for imputation including the three that we developed that can be applied using the PIMMS workflow. Users

can compare the methods, use the imputations and generate summary plots. Supporting so many methods in a general comparison requires the setup to define a common interface, which in our case is a method that imputes the data given only the intensity data on one measurement level.

We considered GSimp, mslImpute, ProJect, MSstats and for inclusion. We managed to add GSimp and mslImpute as they provide either source files we could adapt (GSimp) or installable packages (mslImpute). ProJect provides neither of these and MSstats has no general imputation functionality. In detail:

- GSimp: “GS_impute” provided core functionality via source files which we added to our repository. No instructions were added on installing the needed dependencies and figuring these out for integration into our workflow took hours. We made the effort (PR 55), but with non-packaged code, reproducibility is very hard to achieve. GSimp itself is a combination of QRLIC for initialization with a glmnet model applied iteratively.
- ProJect is not a package (CRAN, BioConductor) nor a single file to source from which we can ship with the workflow. We simply cannot see the algorithm’s performance as it has no documentation or reusable code.
- In MSstats last version, they do not offer a function to just impute. They offer complete workflows that require additional annotations and several levels of data. As the NAGuideR application, they thus require a differential analysis setup to use the tool. Our workflow uses only the intensities, no additional files. The intensities/abundances can represent different levels of aggregation. For the DDA HeLa data, we analyzed the runs individually without match-between-runs or similar. MSstats 4.0 “MaxQtoMSstatsFormat” functionality needs exactly one evidence file next to the protein groups files and experiment annotations. This we do not have for our development data set. For the DIA data which was processed in one go, it requires Spectronaut peptide level output, where we used the protein groups file (PG.Accessions). Furthermore, MSstats does not impute all values, see Lincoln Harris from the Noble lab discussion here. We do not claim that this is not a sensible choice, but we decided to compare general imputation results. We mention the constraint of our workflow in the discussion: Lines

519-520: “Finally, we offer a workflow to reproduce the comparison done here for all 30 general imputation methods using any other single tabular data provided by the user.”

- msImpute has a MAR-focused method ‘v2’ recommended for DIA datasets, and a MNAR-focused method ‘vs-mnar’. The latter needs a group variable, which creates subgroups to run the algorithm if we see this correctly. Therefore we specified only one group indicating to process the data as one batch. msImpute requires at least four observations per feature to run. We updated the MNAR procedure to ensure this as we think it’s a good default minimum. However, this check also indicated to us that on a comparison with high MNAR proportion most measurements of less prevalent features, which more often are from the lower intensity range, will end up in the validation and test split. E.g. on the HeLa development dataset with 50 samples and a share of 75 MNAR in the ten percent simulated missing values, 4 of 36,911 peptides only had less than 4 observations in the training dataset, leaving at least 8 for the other two splits. For these we moved the validation observations to the training data, leaving the test split untouched.

In summary, GSimp only ran for the HeLa dataset with 50 samples (the smaller development dataset). It did not overall show good performance but got better with a higher share of MNAR - which is not surprising given the QRILC initialization. However, also with the highest share of simulated MNAR, it did not rank among the best. Both msImpute implemented methods could run, although it took hours on the larger dataset (**Supp. Data 4**). However, the results on a dataset with many samples are roughly 10 times worse than using median imputation (Supp. Data 3). If nonetheless required, specified methods can be analyzed in detail as done in **Fig. 4c**, where we e.g. added QRILC.

2- Application scope:

In my first review, I insisted on the need to test the PIMMS methods in realistic scenarios, with a varying proportion of MNARs. The authors declined my recommendation using dubious arguments (see next section). On the other hand, the material they present gives evidence their methods are MCAR only. Notably, the observed trend of imputing LOD MVs nearer to the distribution mode in a "conservative" way (see lines 476-478) is a feature common to most of MCAR methods. Similarly, the construction of the train/test split is in general associated to MCAR imputation (which is also the reason why RF works well essentially on MCARs). In itself, proposing an MCAR-only algorithm is not a problem, however:

- MCAR values are the easiest ones to impute and many methods already do so.
- The article never explicitly points this restriction to MCARs, while on the contrary, it makes a big deal of discussing the MAR/MNAR dichotomy, in the introduction, in the results and in the discussion.

Sometimes, this argument is also used (see line 192) to discard imputation methods from comparisons, like RSN (there are datasets on which simple MNAR methods like RSN or other work well, notably for biomarker discovery). Worse, in sentences like those of lines 476-478, the authors present a general feature of MCAR methods as if it was specific to their approach, which is misleading.

In our first revision, we tried to show performance on simulated missing values across the intensity range of the feature they originate from, using the median to describe the distribution (**Fig. 2e,f, Fig. 4c**). We acknowledge the limitations of not testing MNAR in our manuscript and have now redone all relevant analyses using a share of 25 percent MNAR. We used the approach of (Lazar et al. 2016) and initially

tested MNAR at 25, 50 and 75 percent on the development dataset. These results are presented in **Supp. Data 3**. We chose 25 percent MNAR for the rest of the analyses because the overall ranking between models stayed the same on all three setups, but intensities from lower intensity features were sufficiently overrepresented using a share of 25 percent MNAR without removing too many from the training data. Concerning the workflow, users can now set a MNAR proportion in the simulated missing values.

The following paragraphs, figures and tables have been updated to reflect the addition of 25 MNAR, with other paragraphs being affected as a consequence:

Lines 142-169: “Evaluating self-supervised models for imputation of MS data”

Lines 179-232: “Imputing precursors, aggregated peptides, and protein group data”

Lines 303-319: “Performance of PIMMS on simulated missing values in real use case”

Lines 320-341: “Recovering lost differentially abundant protein groups”

Lines 430-452: “Robustness of differently abundant protein group identification”

Figure 2, Figure 3, Figure 4.

Supp. Fig. 4, 5, 9 (and Supp. Fig. 6, 7, 8 and Supp. Table 2 as a consequence)

Supp. Data 3, 5, 7, 8 (and 4, 6, 9, 10, 11 as a consequence)

Overall, the results of our analyses remain the same, showing good results for data-driven models like KNN-based implementation, random forests, IMPSEQ and our introduced deep learning models. BPCA had great performance on the development datasets, however performed poorly on the real-world alcoholic liver disease dataset.

Furthermore, we, initially, only showed the overall best five models’ in the main plots, which is changed now for **Fig. 4c**. There we added QRILC to this Figure to show performance across the range of features.

Finally, in In L476-478 we do not make the alleged claim but stated that the newly introduced models have that capability - we did not say anything about other models:

L476-478 (rev.1): “We therefore argue that our holistic model based imputations are more conservative than e.g. the RSN imputations and that the three self-supervised DL models offer a sensible approach to proteomics imputation while scaling well to high feature dimensions.”

Based on the above we updated the the discussion:

Lines 493-536: “Imputation is an essential step for many analysis types in proteomics, which is often done heuristically. Here we tested three models using a more holistic approach to imputation. We showed that CF, DAE and VAE models reached a similar performance on simulated missing values across

the entire distribution of the data - including low abundant features. In comparison to most other methods they scaled better to high dimensional data or outperformed fast implementations as scikit-learn based KNN, which was especially beneficial when working with peptides instead of protein groups to avoid implicit imputations⁸. Further, we investigated the effect of the imputation method on a concrete analysis, using DIA data from 358 liver patients. Here we found that missing values were imputed by the models towards the lower end of the distribution but less pronounced as when using QRILC or RSN imputation which shifts all replacements towards the limit of detection (LOD) in a sample. We believe that this is due to the lowest abundant features limiting the learned data distributions (**Supp. Fig. S2**) and that some features are not set towards the LOD by the model due to being missing at random. We therefore argue that our holistic model-based imputations are more conservative than e.g. the RSN imputations and that the three self-supervised DL models offer a sensible approach to proteomics imputation while scaling well to high feature dimensions. We found that all methods besides RSN were a better choice for the ALD dataset analyzed.

Simulating 20 percent missing values with a share of 25 percent MNAR we saw the ability to recover signal by the three semi-supervised models. TRKNN and RF recovered lost signal in this application, but median imputation and QRILC were too heuristic to recover most of the signal (**Figure 3**). However, the analysis also revealed that some protein groups will be false positives or negatives as removing some observations can change the outcome. This suggests that imputation using our or other data driven models can recover lost biological signals which is in line with the observation that significant values were not made insignificant when using data driven models (DAE, TRKNN, CF, RF, VAE) in comparison to RSN or QRILC on the ALD cohort (**Figure 4**). Finally, we offer a workflow to reproduce the comparison done here for all 30 general imputation methods using any other single tabular data provided by the user.

A limitation of the model-based approach is that the models should only be used for imputation if the samples are related. Therefore, the best imputation strategy will be dependent on the experimental setup. We showed that the models can learn to perform imputation on plasma samples from a diverse set of clinical phenotypes ranging from healthy to liver cirrhosis. However, the data-driven models would not perform well when imputing for instance one liver proteome together with ten plasma proteomes. In such a case the data-driven models would not have any other liver proteomes to learn from. If replicas on a small number of samples are the study design, KNN interpolation can be a good remedy or using a set of tests which among others capture missingness²³. DAE and VAE are not suited to be trained on too few samples, however the exact cutoff will need to be evaluated on a per dataset basis. CF will by the training design not be too dependent on the number of samples if trained sufficiently. We showed that performance is at least competitive with at least 50 samples. In summary, for highly varying features in a complex experimental setting with many differing samples, a holistic model trained with all features and potentially additional related samples can capture dependencies between features - such as the ones implemented in PIMMS.”

- Line 307 and after, the authors claim they can impute MAR (ie any MV that is not MNAR, not only MCAR ones), while they do not demonstrate it. Unless a specific (not shown) algorithm is used, randomly

removing 20% of the data does not simulate MARs in general, but MCARs. Imputing MAR values with MCAR algorithms is a common shortcut in proteomics as there is no better way, however, it should not lead the authors to overstate their MAR imputation capabilities.

We apologize for not being precise about the different mechanisms of randomness. We now improved the analysis by removing 20 percent of the data with a share of 25 percent MNAR. In this setup, the overall results did not change indicating at least that we can deal with some level of MNAR.

Put together, these elements give the impression the authors overstate their application scope.

3- Biased comparisons and statistically invalid results:

This is probably the most important issue. Although the main text gives the impression the proposed methods perform reasonably well (and often a bit better than their competitors), a deeper look into the supplemental materials tells another story:

- If one looks at NAGuideR's article as well as recent other articles, the 3 best methods seem to be ImpSeq(rob), SeqKNN and BPCA. Conversely, KNN and RF, although efficient on theoretical datasets with MCARs only (see my first review) do not appear to be accurate on real datasets where MARs and MNARs coexist. Knowing this, the prevalence of KNN and RF in the main article is rather puzzling (BPCA and SeqKNN only occasionally appear, as well as MICE-NORM, RSN or MEDIAN).

We did in no way intend to give an inflated performance of the PIMMS models. We now emphasize that at least 50 samples should be present, but that the interesting use cases are clinical studies with hundreds of samples.

In the previous submission and for this one, we selected the five best-performing methods using our evaluation approach, now across the 30 methods included in the workflow - as far as they did run within 24 hours. This also becomes apparent in Supplementary Data 3.

The workflow is dataset focused and there might be differences in what runs where. We are now running it on a Linux machine for the development datasets which improved the number of methods that were able to run. Additionally, we want to mention that for some of the previous comparison papers, for instance, NAGuideR a test split is not used to evaluate the performance. Instead, results are presented on the training data which are prone to overfitting yielding biased results. We provide results on an independent test set which was neither used for training nor model selection.

We added, as mentioned above, a way to oversample less abundant intensities simulating MNAR as in (Lazar et al. 2016). We also use "real datasets" where MAR and MNAR co-exist, but so far we simulated MCAR representing the entire distribution. We changed our default to a share of 25 percent MNAR to have enough statistical power for lower range features. Additionally, we created a way to specify custom model subsets that aim at giving an insight into the performance across the range of feature intensities. Now, we present QRILC next to the best four methods (DAE, TRKNN, CF, VAE) in **Fig. 4c**.

- A similar bias appears in the selection of MNAR-only methods: TrKNN never appears and does not seem to have even been tested according to Supp Data 4 (while it provides rather good performances

according to NAGuideR) and among the many other ones tested (MINIMUM, MINDET, MINPROB, QRILC, ZERO and PI, the latter being equivalent to RSN), QRILC systematically outperforms RSN/PI according to Supp Mat 3, so using RSN as benchmark looks like a convenient bias.

TrKNN is a custom script provided by NAGuideR (which they copied (and adapted?) from the publication) which failed in the previous submission. The installation without a package and the specified dependencies is hard in an automated workflow, especially as we don't want to take care of maintaining the code base of other authors' methods. Nonetheless, we looked into the method here again. The NAGuideR function calls it the following way (link):

```
source('Trunc_KNN/Imput_funcs.r')
sim_trKNN_wrapper <- function(data) {
  result <- data %>% as.matrix %>% t %>% imputeKNN(., k=10,
    distance='truncation', perc=0) %>% t
  return(result)
}
df1x <- sim_trKNN_wrapper(t(df1))
df <- as.data.frame(t(df1x))
```

Removing the repeated transposition and dplyr (?) syntax, this should be the same as

```
source('Trunc_KNN/Imput_funcs.r')
df1x <- imputeKNN(as.matrix(df), k=10, distance='truncation', perc=0)
df <- as.data.frame(df1x)
```

This runs now in our current environment and we have included results of TRKNN in the paper. On the ALD data, it is now among the best performing models.

Regarding the use of RSN instead of QRILC, which is indeed the better method: We used PI/RSN as the reference as this was the method that was used in the original ALD paper. We wanted to analyze this choice as an example in our context. As requested, we extended the comparison by including QRILC as the better MNAR method (**Fig. 4c-g, Supp. Data 9-11**). We observe that overall QRILC shifts the distribution of the imputed values more drastically than RSN (**Fig. 4 d-g, Supp. Fig. 7**).

- In Supp Data 3 (lower table), it appears that with experiments of small to intermediate sizes ($n < 50$), ImpSeq, SeqKNN and BPCA outperform the proposed methods and the other NAGuideR methods. In this context, lines 208-214 are so misleading they can be considered as statistical malpractice. Notably, sentence like "For example with 10 samples the precursors were fitted best with an MAE of 0.57 using BPCA and 0.61 using CF" conveniently hide that in the first column of the table (ProteinGroups_N10), the best PIMMS method displays an MAE 25% larger than ImpSeq. Similarly, in the 4th and 5th columns, the best PIMMS method is not in the Top 3. Considering these numbers, calling any of the PIMMS method competitive requires an important mind twist.

We emphasize that we did not want to hide the results. We also emphasize that the workflow is dataset focused, i.e. the best methods have to be determined per dataset, potentially with many operating

similarly well. We highlighted in our introduction that we developed our methods for medium to large scale dataset:

Lines 128-131: “Here we used large (N≈450) and smaller (N≈50) MS-based proteomics datasets of HeLa cell line tryptic lysates acquired on a single machine (Q Executive HF-X Orbitrap) over a period of roughly two years to evaluate the general performance on repeated measurements of an homogenous biological sample on a medium to large-scale dataset.”

The cited sentence was taken out of context and was followed by the summary where we highlighted that we focused on competitive performance for as few as 50 samples:

Ln 206-214 (rev. 1): “We found that BPCA (Oba et al. 2003) performed best on the 50 homogenous HeLa samples with relatively long runtime (e.g. on precursors: BPAC: 20:44, MICE-NORM: 4:42). For 10, 20, 30 and 40 samples, other methods showed good results, although PIMMS models still fit the data (Supp. Data 3): For example with 10 samples the precursors were fitted best with an MAE of 0.57 using BPCA and 0.61 using CF. In summary, this indicated that on an unnormalized, intensity varying dataset from a single machine the PIMMS models were able to capture patterns between detected features to impute values (**Supp. Fig. S3a**) for as few as 50 samples with competitive performance to other state-of-the-art methods.”

Therefore, we did not intend to hide the performance of the other methods and elaborate on the comparison - only that you might find good results for your given dataset at hand also with the newly introduced models. But this will of course depend on the dataset at hand. To make this even clearer we updated the sentences:

L222-233: “Reducing the training sample by roughly a tenth, performance on the smaller development dataset (N=50) was comparable to the performance on the larger development dataset, while runtime reduced by 5 to 15 times for precursors (Median: 0:23, KNN: 0:33, VAE: 1:04, DAE: 0:52, CF: 1:06). On the medium sized HeLa datasets all models completed (**Supp. Data. 4**) for the protein groups and 26 out of 27 for the peptides and precursors. We found that IMPSEQ and BPCA³⁹ performed best on the 50 homogenous HeLa samples although some with longer runtimes compared to PIMMS models (e.g. on precursors: IMPSEQ: 1:09, BPAC: 20:44, MICE-NORM: 4:42). For fewer than 50 samples the PIMMS models can fit the data, but alternatives were better suited (**Supp. Data 3**). In summary, this indicated that on an unnormalized, intensity varying dataset (**Supp. Fig. S3a**) from a single machine the PIMMS models were able to capture patterns between detected features to impute values for as few as 50 samples with competitive performance to other state-of-the-art methods.”

- Likewise, on the upper table of Supp Data 3, ImpSeq has almost never been tested, while it is computationally rather efficient (at least far more than BPCA), which raises questions.

We found ImpSeq to not be robust in fitting the data. Below is the most common error where a solver runs into a condition of singularity (so it cannot handle low-rank matrices or invert the data matrix). Given that proteomics data is highly correlated, this is unfortunate.

```
Error in solve.default(icovx[mvar, mvar]): system is computationally
singular: reciprocal condition number = 0
Traceback:
```

```
1. nafunctions(df, method)
2. impSeq(df1) # at line 83 of file <text>
3. solve(icovx[mvar, mvar])
4. solve.default(icovx[mvar, mvar])
```

We are not familiar with the implementation details of both algorithms and cannot say if they normally support a dataset of the requested size or with the given properties. We now updated from R 3.6 to R 4.1 or higher (depending on the OS). The updated version runs now at least on the smaller development datasets. Both BPCA and IMPSEQ imputation are available in PIMMS and results are included in the manuscript. We found that both ImpSeq and BPCA were the best methods for our smaller development datasets and that BPCA was slightly better than the semi-supervised models (0.53 vs. 0.54 MAE) on the protein group data from the big development dataset. This is now reflected in an updated **Figure 2** and main text:

L221-232: Reducing the training sample by roughly a tenth, performance on the smaller development dataset (N=50) was comparable to the performance on the larger development dataset, while runtime reduced by 5 to 15 times for precursors (Median: 0:23, KNN: 0:33, VAE: 1:04, DAE: 0:52, CF: 1:06). On the medium sized HeLa datasets all models completed (**Supp. Data. 4**) for the protein groups and 26 out of 27 for the peptides and precursors. We found that IMPSEQ and BPCA (Oba et al. 2003) performed best on the 50 homogenous HeLa samples although some with longer runtimes compared to PIMMS models (e.g. on precursors: IMPSEQ: 1:09, BPAC: 20:44, MICE-NORM: 4:42). For fewer than 40 to 50 samples PIMMS model can fit the data, but alternatives were better suited (**Supp. Data 3**). In summary, this indicated that on an unnormalized, intensity varying dataset (**Supp. Fig. S3a**) from a single machine the PIMMS models were able to capture patterns between detected features to impute values for as few as 50 samples with competitive performance to other state-of-the-art methods.

On the ALD clinical dataset, both ImpSeq and BPCA produced no results (error above) or poor results. For instance, the best-performing models had MAEs of 0.52-0.55 whereas BPCA had an MAE of 2.95. This is now written in the main text:

Ln 307-313: “After imputation we again compared how well PIMMS imputed simulated missing values with a share of 25 percent MNAR in the ALD data and found that DAE, TRKNN, CF, RF and VAE achieved similar results of MAE between 0.52-0.55 (**Fig. 3a, Fig. 4a-c, Supp. Data 3**). BPCA which performed well

on the development dataset, was worse with an MAE of 2.95 and took more than 10 minutes to complete compared to 30 seconds up to just above one minute for the best models (Supp. Data 3, 4).”

- Even if on the largest HeLa dataset, state of the art methods may not scale up, the BPCA performances on the smaller dataset are such that a good imputation strategy would be to apply it batch-wise (in other words, it reaches similar performances than the PIMMS methods with fewer samples). I understand the scalability argument is of interest, but throwing away any method that cannot be fast on a set of 500 samples is a bit excessive. Let us recall that in their original submission, the authors used interpolation as a reference method, which is a KNN-like algorithm applied to a batch of size 3. Likewise, why not comparing state-of-the-art methods (notably BPCA or ImpSeq) on 10 batches of 50 samples? (assuming ImpSeq can indeed not scale, a fact I remain very skeptical about).

The interpolation of samples was done on the dataset loaded into memory and replaced with KNN as the KNN is a generalized (and better) version of the idea. Interpolation on the HeLa samples worked somehow as the same cell line was repeatedly measured, but it failed on a diverse biological dataset with samples from patients. Exploring how to batch a sample, would require to ensure that the workflow behaves differently for different methods. Some methods, e.g. the three models we implemented or KNN based implementation, would use all samples in one batch, whereas others would create their own sub-batches. If the packages (BPCA or ImpSeq) implement this procedure in the future as a callable, it will be easy to integrate it into our workflow. But for now, this would mean implementing our own subroutines in R which is in our opinion out of scope.

We opted for running the packages for an extended amount of time on a machine with many cores and extended memory (40 cores, 192 GB memory). Each method was allocated one core and up to 192 GB of RAM, and up to 24 h of runtime.

- Line 640, The KNN algorithm has not been used with the default tuning in most proteomic applications (K=10), but with K=3, which is likely to artificially lower its performances. Conversely, it appears the HeLa dataset has been used for hyperparameter tuning (Supp Fig 1) as well as for evaluation. Owing to the medium-only stability of DAE and VAE (Supp Fig 1), I fear data overfitting (already mentioned by another reviewer) has artificially boosted the performances displayed. As a result, I suspect the performances displayed are optimistic for the PIMMS methods and pessimistic for the reference methods (notably KNN).

We used K=3 as this is the most common number for technical replicates which we based our initial interpolation approach on (Poulos et al. 2020). Additionally, others used this as a default in their work (Santos et al. 2022). We therefore used this as the default using the scikit-learn implementation. The KNN imputation based on an R package was run as in NAGuideR with K=10, which is already present in the result, so both for TrKNN and KNN_IMPUTE method.

We now additionally tried different K for the scikit-learn based KNNImputer implementation. We ran the workflow for a large protein groups’ HeLa dataset where K=3, K=5, K=10, K=15 are compared against each other (Figure below).

On the large HeLa protein groups dataset a) indicates that the best performance is obtained for five neighbors on b) using a share of 25 percent MNAR in the simulated missing values. The performance is worse on simulated intensities from the lower intensity features, as shown in c). This analysis led us to update the way the workflow can be configured, now allowing several configurations for one model to be compared. Note that this will lead to different colors for the same type of model.

Finally, all our results are on the test split which was not used for training or validation. Therefore, we are confident that our results are not due to overfitting.

- In most of the panels from figures 2 and 3, the authors compare the 3 PIMMS methods to 3 (or less) state of the art imputation methods (generally KNN and RF). In those figures, the best PIMMS method is not always the same (eg it is DAE on 2A and CF on 2B), which can make sense. However, claiming the efficiency of PIMMS approach in general (ie including the 3 different algorithms) as done line 214 or in the discussion (lines 466, 476, 484, etc.) is not correct, as doing so would require adding a multiple test correction. Otherwise, one just has to propose a toolbox with 97 random methods, to compare them with 3 reference ones, and it will be easy to claim that in 97% of the case, one of the proposed method outperforms the reference methods. Therefore, in the current context, the performances are exhibited in a statistically invalid way.

We selected the best N overall methods for each of the analyses defaulting to the best five. In Figure 2 where we combine the best five models over the three datasets, we now show the performance of all models in the entire set if available. All the data used for plotting was and is still available in the supplementary data. Additionally, users can now specify a list of models for custom comparison in case the models are not among the best models. In Figure 4c we showcased this by adding QRILC to the plot. Therefore, we ran all state-of-the-art methods in our comparison but focused on the visualization of the best per dataset.

- Dubious arguments can also be found in the response to reviewers. Notably, claiming a varying ratio of MCARs/MNARs would penalize ML-based models is nonsensical (recall that KNN or RF are also based on training, and such imputation tools have long been evaluated that way). Similarly, assuming binning the MAE according to the intensity (Fig. 2E, 2F, 4C) is of interest to assess the MCAR/MNAR imputation capability does not hold for many reasons, the most blatant one being that in those figures, no MNAR method (like RSN) is compared to!

The workflow figures showed the results for the best five performing methods, where MNAR methods were not present as they performed poorly on the simulated missing data, which was initially MCAR. As QRILC was suggested as a better MNAR method than RSN, we added it to the comparison in Fig. 4c), which users can now do as well.

We agree to disagree on the assessment of oversampling low abundant features on model performance (see above). KNN and RF will also be penalized for learning not so frequent features if intensities from a feature are more present in the validation and test dataset than in the training data set. Furthermore, the MCAR/MNAR in e.g. (Lazar et al. 2016) is done only on a complete data matrix, effectively this would mean in our context to remove all low abundant features upfront and then evaluating MNAR on highly abundant features, e.g. protein groups. Nonetheless, we now replaced our analyses with a share of 25 percent MNAR in the simulated missing data as described in Lazar.

Overall, the amount of omissions, distortions, statistical approximations and biases is so important that the main text seems to describe an "alternative truth" with respect to the numbers and statistics available in the supp mat or in the literature.

This is a strong overreaction to our response, where we tried to argue fairly for our point of view. We would appreciate a more balanced discussion.

Conclusions and recommendations:

Although the originality of the approach is worthwhile, the article cannot be published in its current form, because it steps away from scientific publication good practices (acknowledgement of prior literature, no overstatement/unsupported claim and most importantly, fair comparisons and statistically valid performance assessment). If the authors correct the manuscript accordingly, I think it will be publishable, but I fear it will no longer meet the expectations of a journal like Nat Comm. More precisely, I suspect a fair presentation of the results will show that the 3 presented algorithms are able to impute MCAR-only MVs on very large-only datasets with acceptable-only performances, ie not as good as ImpSeq, SeqKNN, BPCA or msImpute. Nevertheless, their easiness to scale-up may be of interest for some specific applications, so that a publication in a 2nd tier journal would make sense.

References

Lazar, Cosmin, Laurent Gatto, Myriam Ferro, Christophe Bruley, and Thomas Burger. 2016. "Accounting for the Multiple Natures of Missing Values in Label-Free Quantitative

- Proteomics Data Sets to Compare Imputation Strategies.” *Journal of Proteome Research* 15 (4): 1116–25.
- Oba, Shigeyuki, Masa-Aki Sato, Ichiro Takemasa, Morito Monden, Ken-Ichi Matsubara, and Shin Ishii. 2003. “A Bayesian Missing Value Estimation Method for Gene Expression Profile Data.” *Bioinformatics* 19 (16): 2088–96.
- Poulos, Rebecca C., Peter G. Hains, Rohan Shah, Natasha Lucas, Dylan Xavier, Srikanth S. Manda, Asim Anees, et al. 2020. “Strategies to Enable Large-Scale Proteomics for Reproducible Research.” *Nature Communications* 11 (1): 1–13.
- Santos, Alberto, Ana R. Colaço, Annelaura B. Nielsen, Lili Niu, Maximilian Strauss, Philipp E. Geyer, Fabian Coscia, et al. 2022. “A Knowledge Graph to Interpret Clinical Proteomics Data.” *Nature Biotechnology* 40 (5): 692–702.

Reviewers' Comments:

Reviewer #3:

Remarks to the Author:

In their second revision, the authors conducted a significant work. I acknowledge the quality of the revision (see below), but there is still one fundamental issue (see below).

First, I positively notice that all the overstatements and unsupported claims of the manuscript have been smoothed. The performances of the proposed methods are factually reported. The authors fairly acknowledge that their methods do not outperform the state-of-the-art but perform equally well. Yet, their computational speed and scalability to larger datasets are features, which could interest some readers.

Second, on some aspects, the authors went further than I asked for (notably their implementation of so many algorithms from the literature), which must also be acknowledged. This impressive amount of work and the extensive comparisons did a lot to convince me that the unsupported claims of the previous revision were probably caused by an insufficient statistical culture only (see below).

Despite these positive elements, I have one concern of importance: To evaluate the performances with 25% of MNARs, the authors trained their methods using a similar MNAR rate. Thus, they only show their methods can impute MNARs if the MNAR rate is specified and if the MNAR distribution is known, which contrarily to what is claimed does not make the training harder, but easier (as the training and testing data are then iid, as opposed to real life cases). Although other ML-based methods (like RF) are fairly evaluated using the same set-up, the results are non-informative, as on real data no one is able to specify the MNAR rate/distribution. Thus the evaluation set-up artificially promotes all ML-based methods, including PIMMS ones, wrt others. It reminds me of the previous authors' response that realistic evaluations are detrimental to ML-based approaches. However, it is not the problem that adapts the solution's requirements, but the other way around.

It is my third review of this paper. In my two first reviews, I had to argue tightly to obtain comparisons with recent methods from the literature (normally, a must-have preliminary to the submission) as well as unbiased experimental comparisons. I first interpreted the authors' rebuttals and dodging as a paper-pushing attitude, but their last submission (transparent and rigorous, despite a remaining bias about MNARs) made me revise my judgement. I am now under the impression that the problem has lied in the lack of expertise in statistics (as opposed to proteomics or deep learning). Therefore, I feel in an in-between position, where my reviews do not only assess the work accomplished but also provide statistical guidance that should have been provided by the senior authors. It is extremely time-consuming and not so rewarding, so I will not review future versions of this article.

As for editorial advice, I am in-between too: Overall, the seminal idea is appealing, the methods provide acceptable performances in addition to novel computational features (speed and scalability) and the code provided is noteworthy (accessible package, framework for exhaustive comparisons with other methods, etc.). All these deserve publication. On the other hand, publishing a biased experimental protocol, which artificially boosts the performances by giving access to unknown information would open the Pandora box, especially in a flagship journal like Nature Communications. Indeed, any new proposal about imputation would refer to this article in the future to adopt a similar biased evaluation, and proteomic researchers will be deceived by the real-life performances. This in-between makes me believe that adding a 10 lines paragraph in the discussion in order to detail the evaluation bias (and possibly suggest future investigations in estimating the MNAR rate/distribution?) would make the article acceptable for publication, at least in a less influential journal.

REVIEWER COMMENTS

Reviewer #3 (Remarks to the Author):

In their second revision, the authors conducted a significant work. I acknowledge the quality of the revision (see below), but there is still one fundamental issue (see below).

First, I positively notice that all the overstatements and unsupported claims of the manuscript have been smoothed. The performances of the proposed methods are factually reported. The authors fairly acknowledge that their methods do not outperform the state-of-the-art but perform equally well. Yet, their computational speed and scalability to larger datasets are features, which could interest some readers.

Second, on some aspects, the authors went further than I asked for (notably their implementation of so many algorithms from the literature), which must also be acknowledged. This impressive amount of work and the extensive comparisons did a lot to convince me that the unsupported claims of the previous revision were probably caused by an insufficient statistical culture only (see below).

Despite these positive elements, I have one concern of importance: To evaluate the performances with 25% of MNARs, the authors trained their methods using a similar MNAR rate. Thus, they only show their methods can impute MNARs if the MNAR rate is specified and if the MNAR distribution is known, which contrarily to what is claimed does not make the training harder, but easier (as the training and testing data are then iid, as opposed to real life cases). Although other ML-based methods (like RF) are fairly evaluated using the same set-up, the results are non-informative, as on real data no one is able to specify the MNAR rate/distribution. Thus the evaluation set-up artificially promotes all ML-based methods, including PIMMS ones, wrt others. It reminds me of the previous authors' response that realistic evaluations are detrimental to ML-based approaches. However, it is not the problem that adapts the solution's requirements, but the other way around.

It is my third review of this paper. In my two first reviews, I had to argue tightly to obtain comparisons with recent methods from the literature (normally, a must-have preliminary to the submission) as well as unbiased experimental comparisons. I first interpreted the authors' rebuttals and dodging as a paper-pushing attitude, but their last submission (transparent and rigorous, despite a remaining bias about MNARs) made me revise my judgement. I am now under the impression that the problem has lied in the lack of expertise in statistics (as opposed to proteomics or deep learning). Therefore, I feel in an

in-between position, where my reviews do not only assess the work accomplished but also provide statistical guidance that should have been provided by the senior authors. It is extremely time-consuming and not so rewarding, so I will not review future versions of this article.

As for editorial advice, I am in-between too: Overall, the seminal idea is appealing, the methods provide acceptable performances in addition to novel computational features (speed and scalability) and the code provided is noteworthy (accessible package, framework for exhaustive comparisons with other methods, etc.). All these deserve publication. On the other hand, publishing a biased experimental protocol, which artificially boosts the performances by giving access to unknown information would open the Pandora box, especially in a flagship journal like Nature Communications. Indeed, any new proposal about imputation would refer to this article in the future to adopt a similar biased evaluation, and proteomic researchers will be deceived by the real-life performances. This in-between makes me believe that adding a 10 lines paragraph in the discussion in order to detail the evaluation bias (and possibly suggest future investigations in estimating the MNAR rate/distribution?) would make the article acceptable for publication, at least in a less influential journal.

Thank you for the thoughts and acknowledgments on our work. Related to the point of the same MNAR setup in the training data as in the testing data, we have performed new experiments. These show that our previous results were unaffected by this.

To change the composition of samples to a non identically distributed (iid) setting, we simulated missing values only from a subset of the samples. Then some samples were not changed, while the remaining intensities of the changed ones were still in the training data. This procedure is compatible with all methods which were included up to this point. The distribution of intensities between an imputation run with all samples with simulated values and only a subset can therefore then be compared.

Following this line of thought, we used 20 percent of the samples on the ALD to generate missing values from, making sure these were stratified by disease class (Kleiner score): We sampled 1,076 simulated missing values from 68 unique samples for the validation split and 1,086 simulated missing values from 69 sample for the test split. We left 318 samples as they were. Therefore the training data included all 455 samples with and without simulated missing values - breaking the iid assumptions.

As we reduce the size of the validation and test data, fewer intensities could be used for evaluation. Sampling missing values from all samples yielded 7,395 for both validation and test data in the original setup.

Unsurprisingly, leaving more data for training, led to slightly better performance. Therefore we argue that changing all data with the same procedure will yield worse performance compared to only a subset as particularly more of the low abundant features are available for training. In the case of (TR-) KNN imputation this means that more samples are available for imputing a certain feature. Table of comparison:

	All samples have simulated MVs			Subset of samples has simulated MVs		
	count	mean	std	count	mean	std
DAE	7,395	0.525	0.748	1,086	0.512	0.670
TRKNN	7,395	0.532	0.750	1,086	0.517	0.682
CF	7,395	0.537	0.732	1,086	0.544	0.739
RF	7,395	0.541	0.732	1,086	0.533	0.707
VAE	7,395	0.548	0.747	1,086	0.522	0.689
Median	7,395	0.599	0.776	1,086	0.584	0.758

Table of best five and median comparing training on sample with all having simulated values and the ones with only a subset on the test data.

If we restrict the comparison only to the shared simulated missing values we get slightly smaller differences which highlights that on the same subset of simulated missing values the differences are minimal between both approaches - with slight improvements when training on a bit more data:

	All samples have simulated MVs			Subset of samples has simulated MVs		
	count	mean	std	count	mean	std
DAE	1,086	0.519	0.723	1,086	0.512	0.670
TRKNN	1,086	0.520	0.697	1,086	0.517	0.682
CF	1,086	0.546	0.705	1,086	0.544	0.739
RF	1,086	0.536	0.697	1,086	0.533	0.707
VAE	1,086	0.543	0.717	1,086	0.522	0.689
Median	1,086	0.588	0.762	1,086	0.584	0.758

Table of best five and median comparing training on sample with all having simulated values and the ones with only a subset on the test data, compared only on the shared simulated missing values.

We included the full list of results into **Supp. Data 8** and added a sentence to the main text

Lines 316-321: “Having many samples available, we also tested to only simulate missing values on a stratified subset of samples (by Kleiner score, see Methods) leading to 68 samples and 69 samples assigned to the validation and test split. Sampling simulated missing values only from

these led to fewer available intensities for comparison but matched the results when sampling from all available samples missing values (Supp. Data. 8).”

And the following to the discussion:

Lines 556-558: “In general, community-curated benchmarks including datasets and detailed metrics should be discussed by creating one or more ProteoBench modules in a community effort (EuBIC 2024).”